

# The TOMCAT global chemical transport model: Description of chemical mechanism and model evaluation

Sarah A. Monks[1,2,3], Stephen R. Arnold[1], Michael J. Hollaway[1], Richard J. Pope[1,4], Chris Wilson[1,4], Wuhu Feng[1,5], Kathryn M. Emmerson[6], Brian J. Kerridge[7], Barry L. Latter[7], Georgina M. Miles[7], Richard Siddans[7], and Martyn P. Chipperfield[1]

[1]Institute for Climate and Atmospheric Science, University of Leeds, UK.
[2]Chemical Sciences Division, Earth System Research Laboratory, National Oceanic and Atmospheric Administration, Boulder, CO, USA.
[3]Cooperative Institute for Research in Environmental Sciences, University of Colorado, Boulder, CO, USA..
[4]National Centre for Earth Observation, University of Leeds, UK.
[5]National Centre for Atmospheric Science, University of Leeds, UK.
[6]CSIRO Oceans and Atmosphere Flagship, Aspendale, Australia.
[7]Remote Sensing Group, STFC Rutherford Appleton Laboratory, Harwell Oxford, UK.

*Correspondence to:* Sarah Monks (sarah.monks@noaa.gov)

**Abstract.** The TOMCAT 3-D chemical transport model has been updated with the emissions and chemical degradation of ethene, propene, toluene, butane and monoterpenes. The full tropospheric chemical mechanism is described and the model is evaluated against a range of surface, balloon, aircraft and satellite measurements. The model is generally able to capture the main spatial and seasonal features of high and low concentrations of carbon monoxide (CO), ozone ($O_3$), volatile organic compounds (VOCs) and reactive nitrogen. However, model biases are found, some of which are common to chemistry models and some that are specific to TOMCAT and warrant further investigation.

Simulated $O_3$ is found to generally lie within the range of ozonesonde observations and shows good agreement with surface sites. The most notable exceptions to this are during winter at high latitudes, when $O_3$ is underestimated, and during summer over North America, when $O_3$ is overestimated. Global Ozone Monitoring Experiment-2 (GOME-2) comparisons suggest that TOMCAT sub-column tropospheric $O_3$ in DJF may also be underestimated outside of the Arctic, particularly near tropical regions.

TOMCAT CO is negatively biased during winter and spring in the Northern Hemisphere (NH) when compared to ground-based observations and MOPITT (Measurements Of Pollution In The Troposphere) satellite data. In contrast, CO is positively biased throughout the year in the Southern Hemisphere (SH). The negative bias in the NH is a common feature in chemistry models and TOMCAT lies well within the range of biases found in other models, while the TOMCAT SH positive bias is at the upper range of positive biases reported in other models.

Using two simulations with different boundary conditions highlighted the sensitivity of model performance to the chosen emission dataset when simulating VOCs, nitrogen oxides ($NO_x$) and peroxyacetyl nitrate (PAN). VOC measurements show winter/spring negative biases in C2-C3 alkanes and alkenes, which is likely driven by underestimated anthropogenic emissions. TOMCAT is able to capture the seasonal minima and maxima of PAN and $HNO_3$. However, comparisons to an aircraft clima-





tology show that PAN may be overestimated in winter and HNO$_3$ may be overestimated in winter and spring in regions over North America. The model showed different biases in NO$_x$, depending on location, with evidence of underestimated Asian emissions contributing to negative model biases over China and underestimated fire emissions contributing to negative biases in the SH.

5  TOMCAT global mean tropospheric hydroxyl radical (OH) concentrations are higher than estimates inferred from observations of methyl chloroform, but similar to, or lower than, multi-model mean concentrations reported in recent model intercomparison studies. TOMCAT shows peak OH concentrations in the tropical lower troposphere, unlike other models, which show peak concentrations in the tropical upper troposphere. This is likely to affect the lifetime and transport of important trace gases and warrants further investigation.

## 1 Introduction

Atmospheric chemistry plays a central role in air quality and climate change, which can have a negative effect on humans on a global-scale. Air pollution has been estimated to have caused over 3 million deaths worldwide in 2010 and this rate is estimated to double by 2050 due to projected increases in emissions (Lelieveld et al., 2015). Increases in anthropogenic emissions have led to higher atmospheric concentrations of greenhouse gases, such as methane (CH$_4$) and ozone (O$_3$), contributing significantly to the observed rise in global mean surface temperature (Stocker et al., 2013). Chemical processing, emissions, and transport determine the concentrations and distribution of pollutants within the atmosphere and the impact that they have on society. Reactive gases, such as volatile organic compounds (VOCs) and nitrogen oxides (NO$_x$), influence air quality and climate as they result in the formation of O$_3$ and aerosols. Other gases such as carbon monoxide (CO), which may not directly affect the climate, can have secondary impacts by influencing the lifetime of gases such as CH$_4$ (Berntsen et al., 2005).

Atmospheric chemistry models help to inform our understanding of how atmospheric chemistry affects climate and air quality on a global- or regional-scale. These models can be used to simulate the temporal and spatial evolution of important short-lived pollutants, taking into account the main physical and chemical processes that act on trace constituents in the troposphere (emissions, chemistry, transport and deposition). The chemical and dynamical complexity and the spatial resolution of such models is a compromise between model accuracy and computational efficiency. Atmospheric chemistry models are often run as chemical transport models (CTMs), where transport is constrained by reanalysis products that assimilate meteorological observations. This allows the simulated chemical fields to provide context for measurements, which are often limited spatially and temporally. They can also be used to further understand the impacts of new atmospheric processes that have been identified by measurements (e.g., Lelieveld et al., 2008). CTMs are of particular use in investigating the impacts of natural and anthropogenic emissions on atmospheric burdens of pollutants that are important for air quality and climate reasons and for source-receptor studies for policy-making purposes (e.g., Sanderson et al., 2008; Fiore et al., 2009).

The TOMCAT CTM is a three-dimensional (3-D) global Eulerian model that has been used for a wide range of tropospheric and stratospheric chemistry studies. For example, it has been used to investigate the impacts of O$_3$ on crop yields (Hollaway et al., 2012), fire emissions on Arctic interannual variability (Monks et al., 2012) and to identify the main sources of peak





summertime $O_3$ in the Mediterranean (Richards et al., 2013). In the stratosphere the model has been used to study issues such as ozone depletion (e.g., Chipperfield et al., 2015) and the impact of solar variability (e.g., Dhomse et al., 2013). TOMCAT is also the host model for the GLOMAP aerosol module (Mann et al., 2010).

This paper summarises recent updates to the tropospheric chemical mechanism, documenting the current full chemical
scheme (Section 2). Key gas-phase species simulated by the latest version of the model are shown and evaluated using a range of observations. The model simulations that are evaluated are described in Section 2.2 and the observations that are used are described in Section 3. The observational platforms that are used include surface, satellite, aircraft and balloon sounding measurements. The model results and comparisons with observations are shown in Section 4 and focus on annual, seasonal and monthly mean simulated concentrations. The chemical species that are discussed include CO, $O_3$, VOCs, reactive nitrogen
($NO_y$) and the hydroxyl radical (OH).

## 2   The TOMCAT model

The TOMCAT model is an Eulerian offline 3-D global CTM and is described by Chipperfield (2006). The model has a flexible horizontal and vertical resolution and the vertical domain can be varied depending on the problem being studied. Typical horizontal resolutions range from 5.6° x 5.6° for multidecadal stratospheric studies to 1.2° x 1.2° for short case studies. The
model uses a $\sigma$ - $p$ coordinate system, with near-surface levels following the terrain ($\sigma$) and higher levels ($\sim$ >100 hPa) using pressure levels ($p$). The model extends from the surface to $\sim$10 hPa for tropospheric simulations, as used in this study. Model meteorology is forced by winds, temperature and humidity fields from the European Centre for Medium-Range Weather Forecasts (ECMWF) reanalyses (Dee et al., 2011). These data are read in every 6 hours and interpolated to the TOMCAT grid. To avoid inconsistencies between horizontal and vertical winds after this interpolation, the vertical motion is diagnosed from
horizontal divergence instead of using analysed vertical velocities. Large-scale tracer advection in the meridional, zonal and vertical direction is based on the Prather (1986) scheme, which conserves mass and maintains tracer gradients (Chipperfield, 2006). Sub-grid scale transport (boundary layer mixing and convective transport) is treated in the model using the Holtslag and Bolville (1993) and Tiedtke (1989) schemes. There is also an option to run the model using archived convective mass fluxes (Feng et al., 2011). Wilson et al. (2014) used sulphur hexafluoride ($SF_6$) to evaluate model tracer transport and showed that the
model is able to reproduce seasonal transport timescales and patterns along with the location of the intertropical convergence zone. However, they also noted that the model inter-hemispheric transport is somewhat slow, resulting in an interhemispheric gradient in $SF_6$ that was 18% too large.

Natural and anthropogenic emissions are read into the model on a 1°x1° resolution and regridded online to the model grid. The model is usually provided with monthly mean emissions and a temporal interpolation is performed online to the model
time step. Isoprene emissions are emitted with a diurnal cycle imposed online to account for the dependence of emissions on daylight. Lightning emissions of $NO_x$ are coupled to convection in the model and therefore vary in space and time according to the seasonality and spatial pattern of convective activity (Stockwell et al., 1999). Aircraft emissions of $NO_x$ are based on estimated aircraft movements for the year 2002 (Lamarque et al., 2010) and were calculated for the European QUANTIFY project



(http://www.pa.op.dlr.de/quantify/). They are available on 25 vertical levels from the surface to 14.5 km and are regridded to the TOMCAT vertical levels online.

Dry deposition velocities are weighted by prescribed fixed land cover fields and seasonally varying sea-ice fields from
the NCAR community land model (CLM) (Oleson et al., 2010). The 16 CLM land types were regridded onto the model resolution and reclassified into the TOMCAT's five land types (Forest, grass/shrub/crop, bare ground, sea-ice and water). Chemical species' deposition velocities were then determined based upon time of day, season and were weighted by the proportion of the grid box covered by each land type. Wet deposition is parameterised according to the proportionality of the removal rate to the concentration of the species and is dependent on convection rates, precipitation and the solubility of gases.
The scheme has been shown to perform well within the TOMCAT model with a 4% bias compared to Radon observations (Giannakopoulos et al., 1999).

## 2.1 Tropospheric chemistry scheme

TOMCAT $O_X$-$HO_X$–$NO_X$-CO-$CH_4$ and C1-C3 alkane hydrocarbon chemistry was previously described by Arnold et al. (2005). TOMCAT has since been extended to also include the oxidation of isoprene based on the Mainz Isoprene Mechanism
scheme (Pöschl et al., 2000). The implementation of this scheme into TOMCAT is described by Young (2007). Most recently, the TOMCAT model has been updated to include the emission and destruction of some C2-C7 unsaturated and aromatic hydrocarbons (ethene, propene, toluene and butane) based on the Extended Tropospheric Chemistry scheme (ExtTC) (Folberth et al., 2006). Biogenic emission and chemistry of monoterpenes based on the MOZART-3 chemical mechanism (Kinnison et al., 2007) has also been added. The model now includes 79 species and they are listed in Table 2, identifying whether they
are emitted or undergo dry or wet deposition. A few shorter lived species are grouped into families for transport processes, which are also identified in Table 2. The chemical reactions are implemented via a software package, ASAD (Carver et al., 1997). Photolysis rates are calculated online based on Hough (1988), which considers direct and scattered radiation. Within TOMCAT, this scheme is supplied with surface albedo, monthly mean climatological cloud fields and total column ozone and temperature profiles. The bimolecular and termolecular kinetic rates are mostly taken from the International Union of
Pure and Applied Chemistry (Atkinson et al., b) and the Leeds Master Chemical Mechanism (MCM, 2004). The bimolecular, termolecular and photolysis reactions are listed in Tables 3-5.

Heterogeneous chemistry is known to affect the global concentrations of $O_3$, OH and $NO_x$ in the troposphere (Jacob, 2000). One important reaction is that of dinitrogen pentoxide ($N_2O_5$) with water ($H_2O$) on the surface of aerosols to form nitric acid ($HNO_3$). $HNO_3$ is highly soluble and is therefore efficiently lost through wet deposition, making this an important loss channel
for $NO_x$ from the atmosphere. This is important in the troposphere when there is no sunlight, allowing time for the formation of $N_2O_5$. TOMCAT can be run coupled to the GLOMAP aerosol module (Mann et al., 2010), which can then calculate the available aerosol surface area for use in the heterogeneous chemistry calculation (e.g., Breider et al., 2010). When TOMCAT is run without the coupled GLOMAP scheme there is an option to account for heterogeneous uptake of $N_2O_5$ using prescribed monthly mean aerosol concentrations that have been calculated for the year 2000 by the GLOMAP model (Mann et al., 2010). In this simplified scheme, uptake coefficients for sulphate, black carbon, organic carbon and sea salt are based on Evans and



Jacob (2005) and the uptake coefficient for dust is based on Mogili et al. (2006), with the overall uptake coefficient varying as

a function of temperature, humidity and aerosol composition. Similarly, computationally cheap TOMCAT-GLOMAP 'aerosol-only' experiments can be run using specified fields of oxidants.

## 2.2  Model simulations and set-up

Two simulations have been performed using the new chemical mechanism scheme for the year 2000 (RUN_2000) or year 2008 (RUN_2008) (both with a 1-year spin-up), which differ slightly in their inputs. Using two simulations for different years and

with different set-ups allows insight into whether a model bias is systematic or possibly due to the model boundary conditions, such as emissions. Both simulations use 31 vertical levels (surface to 10 hPa) and a horizontal resolution of $2.8° \times 2.8°$. Each model run uses ERA-Interim meteorology and emissions for the specific year of the run and uses offline aerosols for $N_2O_5$ uptake.

RUN_2000 uses anthropogenic and biomass burning emission estimates for the year 2000. The anthropogenic and ship

emissions are from a dataset that was created for the IPCC Fifth Assessment Report (AR5) (Lamarque et al., 2010). The biomass burning emissions are taken from the Global Fire Emissions Database (GFED) version 3.1 (van der Werf et al., 2010). Oceanic CO and soil $NO_x$ emissions are from the POET emission inventory (Granier et al., 2005) and biogenic emissions of volatile organic compounds (VOCs) were calculated offline by the Model of Emissions of Gases and Aerosol from Nature version 2.1 (Guenther et al., 2012) within the NCAR Community Climate Model (Lawrence et al., 2011). The MEGAN

emissions are described and used in Scott et al. (2016). Due to the long lifetime of methane ($\sim$10 years), a very long spin-up time would be required in order to simulate a realistic atmospheric methane distribution using online emissions and loss. To avoid this long spin-up, a common method in CTMs is to used a fixed methane field from offline sources. In this TOMCAT simulation, $CH_4$ emissions are from Wilson et al. (2016), with tropospheric surface concentrations being scaled to match a realistic global mean concentration of 1800 ppbv. This results in realistic model concentrations of $CH_4$, whilst the spatial

distribution of high/low emission regions is maintained.

RUN_2008 uses emissions that were chosen for the POLARCAT (POLar study using Aircraft, Remote Sensing, surface measurements and models of Climate, chemistry, Aerosols, and Transport)) Model Intercomparison Project (POLMIP) (Emmons et al., 2015). Monthly mean anthropogenic and ship emissions are based on the Streets v1.2 inventory (Zhang et al., 2009), which was updated with the latest regional inventories in 2008 for the POLARCAT campaign. Monthly mean biogenic emissions are from the MACC (Monitoring Atmospheric Composition and Climate) project (MACCity), which provides simulated VOCs calculated offline by the Model of Emissions of Gases and Aerosols from Nature (MEGAN) v2.1 (Guenther et al., 2012). Oceanic CO and VOC emissions and soil $NO_x$ are from the POET inventory. For 2008, daily biomass burning emissions are taken from the Fire INventory from NCAR (FINN) (Wiedinmyer et al., 2011). Surface $CH_4$ is set to equal zonal

mean concentrations calculated from NOAA/ESRL/GMD surface observations observations for the year 2000 (Meinshausen et al., 2011).

The emissions for both runs are shown in Table 1. Some differences exist between the two sets of emissions, with RUN_2000 having higher total biogenic emissions and RUN_2008 having higher total anthropogenic and fire emissions.



## 3  Observations

### 3.1  Satellite data

Simulated CO is compared on a global scale to CO distributions retrieved from the satellite instrument, MOPITT (Measurements Of Pollution In The Troposphere) version 6. MOPITT is a nadir-viewing instrument on-board the NASA Terra satellite and retrieves CO concentrations globally at a horizontal resolution of $\sim$22 km by measuring infrared radiances in the CO absorption band (Deeter et al., 2010). The Terra satellite has an overpass time at the equator of 10:30 local time (LT). Version 6 uses an *a priori* based on climatological output from the CAM-Chem model for 2000 to 2009 (Deeter, 2013). It has increased sensitivity to lower tropospheric CO by using both near-infrared and thermal infrared wavelengths (Deeter et al., 2011). As MOPITT is a nadir-viewing instrument, it is more sensitive to certain altitudes, therefore averaging kernels (AKs) that contain information about the instrument's varying sensitivities at different altitudes are used, along with the *a priori*, to apply the same vertical sensitivity to the TOMCAT CO profiles. This allows a more accurate comparison between the observed and simulated CO. Data where the degrees of freedom signal (DOFS) are less than 1 have been removed from the model and satellite columns to identify retrievals where the satellite sensitivity is low.

Satellite $O_3$ is taken from Global Ozone Monitoring Experiment-2 (GOME-2) aboard EUMETSAT's Metop-A polar orbiting satellite. GOME-2 is a nadir-viewing instrument with an approximate local equator crossing time of 09:30 LT. It has a spectral range of 240–790 nm and the pixel sizes are between 40 km and 80 km along and across track, respectively (Miles et al., 2015b). The data comes from the Rutherford Appleton Laboratory and is based on an optimal estimation algorithm (Rodgers, 1976). Miles et al. (2015b) describes how the GOME-2 retrievals are quality controlled prior to use, with data being removed where geometric cloud fraction is greater than 0.2 and the solar zenith angle is less than 80°. For optimal comparisons, the GOME-2 AKs are applied to the TOMCAT data, as described in Miles et al. (2015a).

For nitrogen dioxide ($NO_2$), we use data from the Ozone Monitoring Instrument (OMI) aboard NASA's EOS-Aura polar orbiting satellite. It has an approximate equator crossing of 13:30 LT (Boersma et al., 2007) and is a nadir-viewing instrument with a spectral range of 270–500 nm. The pixel sizes are between 16–23 km and 24–135 km along and across track, respectively, depending on the viewing zenith angle (Boersma et al., 2007). The tropospheric column $NO_2$ data, known as the DOMINO product (v2.0) (Boersma et al., 2011), was downloaded from the Tropospheric Emissions Monitoring Internet Service (TEMIS; http://www.temis.nl/airpollution/no2.html). The retrieval of OMI tropospheric column $NO_2$ is based on Differential Optical Absorption Spectroscopy (DOAS), as discussed by Eskes and Boersma (2003). OMI retrievals have been quality controlled and data is only used where they have geometric cloud cover less than 20 % and good quality data flags. The product also uses the algorithm of Braak (2010) to remove OMI pixels affected by row anomalies. Studies have shown the DOMINO product to have small biases against other independent observational data with some evidence of a small low bias over oceans (Irie et al., 2012; Boersma et al., 2008). The product has also been used in model evaluation studies previously (e.g., Huijnen et al., 2010; Pope et al., 2015). For the TOMCAT comparisons, AKs are applied following Boersma et al. (2011).





## 3.2 Surface data

We take $O_3$ measurements at the surface over the U.S. from the United States Environmental Protection Agency (EPA) Clean Air Status and Trends Network (CASTNET) database. They provide hourly mean concentrations from continuous ozone monitoring instruments that have undergone a large amount of quality assurance. Here we use an average of data available for the years 2000 to 2008 at 44 sites covering the whole of the U.S., excluding highly urbanised sites as identified by Sofen et al. (2016). The model output is interpolated to the location of each station both horizontally and vertically.

Observations of CO, VOCs, peroxyacetyl nitrate (PAN) and some $O_3$ measurements are taken from the World Data Centre for Greenhouse Gases (http://ds.data.jma.go.jp/gmd/wdcgg/; see Figure 1 for locations). Most of the surface $O_3$ and CO measurements are provided by the National Oceanic and Atmospheric Administration (NOAA). NOAA CO is from flask samples that have been analysed using gas chromatography (Novelli et al., 1998) and $O_3$ is measured by ultraviolet (UV) light absorption at 254 nm (Oltmans and Levy, 1994). The $O_3$ measurements at Cape Verde are provided by the University of York and were made using a UV light absorption instrument (Read et al., 2008). CO at Minamitorishima is from continuous measurements made by the Japan Meteorological Agency (JMA) using a gas chromatography (Watanabe et al., 2000). PAN measurements at Zugspitze and Schauinsland are provided to the WDCGG by the German Federal Environment Agency (UBA) and were made using a commercial gas chromatograph (GC) analyser (Pandey Deolal et al., 2014). VOC measurements of ethene, ethane, propene, propane, toluene and butane made using gas chromatography at Hohenpeissenberg were provided by the German Meteorological Service (DWD) (Plass-Dülmer et al., 2002). All $NO_x$ measurements were made using Chemiluminescence and are provided by DWD at Hohenpeissenberg (Mannschreck et al., 2004), UBA at Zugspitze, Empa (Swiss Federal Laboratories for Materials Science and Technology) at Jungfraujoch, Payerne and Rigi (Zellweger et al., 2003), and by RIVM (Netherlands National Institute for Public Health and the Environment) at Kollumerwaard.

## 3.3 Ozonesonde data

Simulated $O_3$ profiles are compared to ozonesonde data from a climatology, which uses 17 years of ozone balloon soundings made between 1995 and 2011 (Tilmes et al., 2012). The data is available as profiles between 1000 hPa and 10 hPa at 42 stations, covering large parts of the globe. The model output is interpolated to the longitude and latitude of each station location. The site locations are shown in Figure 1b. The ozonesondes tend to measure concentrations around 10 ppbv higher over eastern U.S. and around 5 ppbv lower over Europe compared to independent observational data from aircraft and surface data (Tilmes et al., 2012). For comparison to TOMCAT, both the model and the observations have been averaged into 3 different altitude and latitude bands for comparison.

## 3.4 Aircraft climatology

Measurements from several aircraft campaigns have been compiled into regional mean profiles calculated over domains covered by the aircraft (Emmons et al., 2000). The so-called climatology includes campaigns that were conducted between 1983 and 2001, covering several months of the year and both hemispheres, providing an insight into the temporal and spatial distribution



of some species that are not routinely measured. Whilst this climatology is useful for this purpose, it is noted that aircraft campaigns often target pollution plumes that may not be captured by global models, in addition to the model generally not being representative of the year in which the observations were collected. This means that differences in meteorology and emissions may play a role in some of the discrepancies. For the TOMCAT comparisons we limit the data to the campaigns that occurred in the most recent 10 years (1992–2001).

## 4   Results

### 4.1   Simulated distributions of CO, $O_3$ and OH

Figure 2 shows annual mean surface and zonal mean concentrations of CO, $O_3$ and OH from RUN_2008. CO is emitted directly from natural and anthropogenic sources and produced in the atmosphere from chemical destruction of VOCs (Logan et al., 1981). Direct emission at the Earth's surface and secondary production in the troposphere from VOCs (most notably
$CH_4$) are estimated to be of equal importance in terms of total global tropospheric CO sources (Duncan et al., 2007). High concentrations due to direct emission of CO from fossil fuel burning can be seen in Figure 2a in the densely populated regions of North America, Central Europe and Asia. Large concentrations are also seen over regions with high rates of biomass burning, such as South America and Africa. Both at the surface and throughout the troposphere higher background concentrations of CO are seen in the Northern Hemisphere (NH) due to larger emissions.

$O_3$ is important in the troposphere as it is a major source of OH, the primary oxidising agent in the troposphere, and is an air pollutant and greenhouse gas (Monks et al., 2015a). It is not directly emitted but produced from photochemical reactions involving $NO_x$, VOCs and CO, and is transported from the stratosphere to the troposphere (Lelieveld and Dentener, 2000). The atmospheric burden of $O_3$ is controlled by a balance between these sources and loss through chemical reactions and deposition (Stevenson et al., 2006). Figure 2c shows the highest concentrations at the surface lie within the NH extra-tropical region due
to the proximity to large emissions of $NO_x$ and VOCs, and photochemical production. Some of the highest concentrations of $O_3$ are found downwind of regions with high NH anthropogenic emissions (identified by CO in Figure 2a). This is due to production of $O_3$ being greater downwind of source regions away from very high $NO_x$ concentrations that can titrate $O_3$ in urban environments (Monks et al., 2015a). Low $O_3$ over the central Pacific Ocean and northern South America is also seen in the model, most likely due to abundance of water vapour in the tropical regions limiting $O_3$ production through the reaction of $O(^1D)$ with $H_2O$. Zonally averaged $O_3$ in Figure 2d shows the highest concentrations of $O_3$ in stratosphere. Low concentrations are seen at higher altitudes in the tropics due to the uplift of low $O_3$ concentrations within deep convection, whilst transport of stratospheric $O_3$ to the troposphere can be seen in the extra-tropics due to the Brewer-Dobson circulation (Butchart, 2014).
The overall features of TOMCAT $O_3$ are consistent with multi-model results from the Atmospheric Chemistry and Climate Model Intercomparison Project (ACCMIP) (Young et al., 2013) and those observed by satellite (Ziemke et al., 2011). TOMCAT simulates an annual mean tropospheric burden of 336 Tg in RUN_2000 and 331 Tg in RUN_2008, which agrees well with the present day ACCMIP multi-model mean tropospheric ozone burden of 337±23 Tg (Young et al., 2013).



OH is the dominant radical responsible for the removal of pollutants such as $NO_x$ and VOCs from the atmosphere, initiating

the production of $O_3$ (Gligorovski et al., 2015) and aerosols (e.g., Carlton et al., 2009). OH is produced in the troposphere

when $O_3$ is photolysed to produce $O(^1D)$ and subsequent reaction with $H_2O$. It is therefore produced in large quantities in the

tropics, where there are large concentrations of $H_2O$ and a high incidence of solar radiation. This can be seen in TOMCAT in

Figure 2e-f with high concentrations of OH occuring between 50°N and 50°S. The spatial distribution of TOMCAT surface

OH is broadly similar to multi-model surface OH from the ACCMIP study shown by Voulgarakis et al. (2013).

## 15   4.2   Evaluation of OH

OH is difficult to measure due to its very short lifetime (∼1s) and low concentrations, and even with vast improvements to *insitu*

measurement techniques (Heard and Pilling, 2003), they do not provide a global picture. A common method to estimate OH is

by using measurements of methyl chloroform ($CH_3CCl_3$, MCF), for which the primary loss channel is through reaction with

OH. Accurate determination of OH from MCF relies on accurate estimation of emissions and the use of models, introducing

possible biases. These measurements are frequently used to estimate the global mean OH concentration (e.g., Krol et al.,

1998; Prinn et al., 2001; Montzka et al., 2011) and can offer some insight into the regional distribution of OH (e.g., Krol and

Lelieveld, 2003; Patra et al., 2014).

TOMCAT global mean airmass-weighted tropospheric OH was calculated using a climatological tropopause (see definition

in Figure 3) following Lawrence et al. (2001). TOMCAT has an annual mean tropospheric OH concentration of $1.09 \times 10^6$

molecules/cm$^3$ in RUN_2000 and $1.08 \times 10^6$ molecules/cm$^3$ in RUN_2008. Concentrations of global mean tropospheric OH

calculated from MCF observations have been estimated to be $0.94 \times 10^6$ molecules/cm$^3$ by Prinn et al. (2001), $1.0 \times 10^6$

molecules/cm$^3$ by Krol et al. (2003) and $0.98 \times 10^6$ molecules/cm$^3$ by Bousquet et al. (2005). These estimates indicate that

the TOMCAT global mean OH may be slightly high. However, a recent inverse modelling study calculated a global mean OH

concentration of $1.06 \times 10^6$ molecules/cm$^3$, highlighting uncertainties in using MCF observations to calculate OH (Wang et al.,

2008). In addition to this, concentrations reported by model intercomparison studies are also higher than those reported based

on observations. The POLARCAT Model Intercomparison Project (POLMIP) found a multi-model mean value of $1.08\pm0.6$

$\times 10^6$ molecules/cm$^3$ when using 8 models (including a previous version of TOMCAT). The multi-model mean was the same

whether a climatological tropopause was used, as done here, or when the 150 ppb $O_3$ contour line was used. Voulgarakis

et al. (2013) found a multi-model mean concentration of $1.17\pm0.1 \times 10^6$ molecules/cm$^3$, when using a subset of 12 ACCMIP

models, and Naik et al. (2013) found a multi-model mean of $1.11\pm0.2 \times 10^6$ molecules/cm$^3$, when using all 16 ACCMIP

models. Both of these ACCMIP concentrations were calculated using a tropopause of 200 hPa. However, Voulgarakis et al.

(2013) found little difference in the resulting concentrations of OH when using different methods of defining the tropopause

(200 hPa, 150 ppbv $O_3$ contour and the climatological tropopause, as used here).

Whilst comparing the global mean OH concentration in TOMCAT to those reported in the literature is very useful, it is also

important to consider the regional distribution of OH in TOMCAT. Figure 3 shows TOMCAT OH from both runs averaged

into 9 regional subsections defined by Lawrence et al. (2001), along with OH from Spivakovsky et al. (2000) (referred to as the

Spivakovsky dataset) and the multi-model mean OH from the ACCMIP study (Naik et al., 2013). Patra et al. (2011) used the





Spivakovsky dataset in a recent multi-model intercomparison project, but revised the concentrations down by 8% to match more
recent measurements of MCF. This highlights that quantitative comparison of TOMCAT OH with the Spivakovsky dataset is
limited due to observational and modelling uncertainties. However, the Spivakovsky dataset is still valuable for estimating the
regional distribution of OH.

The largest concentrations of OH are found in the tropics for the Spivakovsky dataset and for the ACCMIP and TOMCAT
simulations. However, the ACCMIP models have the highest OH concentrations between 500 hPa and 250 hPa, Spivakovsky
has the highest concentrations between 750 hPa and 500 hPa and TOMCAT has the highest concentrations between the surface
and 750 hPa. Large differences in the spatial distribution of simulated OH has recently been identified in models, highlighting
uncertainties in the ability of current models to accurately simulate OH concentrations and distributions (Emmons et al., 2015;
Monks et al., 2015b). TOMCAT was shown to have lower photolysis rates in the upper troposphere and higher photolysis rates
in the lower troposphere compared to other models, with model differences in clouds and water vapour in the POLMIP models
being identified as possible reasons for differences in the OH (Emmons et al., 2015; Monks et al., 2015b).

In addition to this, Patra et al. (2014) found that the NH to SH ratio of OH, inferred from observations of MCF, is equal to
0.97. TOMCAT has an annual NH/SH ratio of 1.37 in both runs. Naik et al. (2013) found a NH/SH ratio of 1.28±0.1 for the
ACCMIP models, which is also higher than that estimated from observations, indicating that this is a common feature in global
models.

TOMCAT OH results in a chemical methane lifetime of 7.6 yrs in RUN_2000 and 7.9 yrs in RUN_2008. Voulgarakis et al.
(2013) found an ACCMIP multi-model mean methane lifetime of 9.3±0.9, with a minimum of 7.1 years and a maximum of
13.9 years. This indicates TOMCAT has a methane lifetime that is generally shorter than other models. As the majority of
methane oxidation occurs in the tropics near the surface (Lawrence et al., 2001; Bloss et al., 2005), the short methane lifetime
is likely due to TOMCAT having a higher concentration of OH in this region compared to other models.

**4.3    Evaluation of carbon monoxide**

As mentioned in Section 4.1, CO is emitted from a wide range of natural and anthropogenic sources and can provide insight
into model emissions and subsequent transport of sources due to its lifetime of several months. Figure 4 shows retrieved CO
from MOPITT (see Section 3) at 500 hPa during April and October 2008 along with simulated CO from RUN_2008 with the
MOPITT averaging kernels applied.

In April, both the model and the satellite show higher CO concentrations in the NH compared to the Southern Hemisphere
(SH) due to a longer CO lifetime at this time of year in conjunction with higher anthropogenic emissions in the NH. MOPITT
observes concentrations around 10–30 ppbv larger than simulated in the NH mid-latitudes and Arctic (Figure 4c). This negative
model bias is a well-known problem with current CTMs during winter and spring, with models having a 15 to 50 ppbv negative
bias against MOPITT at 500 hPa in April in the NH (Shindell et al., 2006) and 5 to 40 ppbv negative bias against Arctic surface
stations in Spring (Monks et al., 2015b). The model shows the best agreement in the NH tropics at this time of year.

TOMCAT CO concentrations in the SH in April are around 10–15 ppbv larger than observed. Shindell et al. (2006) found
good agreement between a 26-model ensemble mean at 500 hPa compared to MOPITT, with individual models showing both



negative and positive biases of between -15 ppbv and +15 ppbv, showing that the TOMCAT bias at this time of year is at the
high end of the multi-model positive bias range.

The model negative bias in the NH and positive bias in the SH leads to a simulated interhemispheric gradient that is too low
(see Figure 4c), which is a common feature in chemistry models (Shindell et al., 2006). Several inverse modelling studies have
suggested that wintertime CO emissions in the NH need to be increased in order to better match observations of CO (Pétron
et al., 2004; Kopacz et al., 2010; Fortems-Cheiney et al., 2011). In addition to this, as mentioned in Section 4.2, the NH/SH
OH ratio in TOMCAT and other chemistry models is higher than estimates based on observations, suggesting that there is too
much OH in the NH or too little in the SH. This is likely to influence the lifetime of simulated CO and will contribute to the NH
and SH biases. Strode et al. (2015) showed that by lowering the NH/SH OH ratio of current state-of-the-art models, simulation
of CO can be improved. The cause of the lower simulated NH/SH OH ratio in models is still unclear and may be linked to
emission biases, where higher emissions of CO and VOCs in the NH may reduce OH concentrations, reducing the NH/SH OH
ratio.

In October, the interhemispheric gradient in CO is no longer as clear due to longer CO lifetimes in the SH and shorter life-
times in the NH. This time of year is characterised by peak fire emissions in the SH (van der Werf et al., 2010). For this reason,
high concentrations of CO are seen by MOPITT over South America and there is a shift in the biomass burning emissions
further south over Africa, resulting in higher CO over the Southern Ocean. TOMCAT also shows higher concentrations over
the Southern Ocean due to the influence of fire emissions compared to April. However, fire emission location errors are clearly
contributing to a mismatch between the CO plumes in the model and those seen by MOPITT. Total column CO over this region
suggests that emissions from fires may be too large in the tropics, particularly over tropical Asia (not shown), and the fires are
located too far north in Africa and too far west in South America, resulting in too much CO being transported out over the
oceans in the tropics (see Figure 4d,e). Naik et al. (2013) also showed that the ACCMIP multi-model annual mean simulated
CO at 500 hPa was 2–45 ppbv too high compared to MOPITT in this region supporting a high bias in CO fire emissions across
different emission inventories in the SH and tropics at this time of year. Outside of the $10°S–30°N$ region, the zonal mean CO
shows much better agreement between TOMCAT and MOPITT than seen in April (see Figure 4f).

Figure 5 compares simulated and measured CO at 14 different surface observatories that are located at several different
latitudes and longitudes (see Figure 1a for station locations). TOMCAT generally captures the seasonal cycle, with high cor-
relations values found at most stations (see r values in Fig. 5). However, the amplitude of the seasonal cycle is often less
pronounced in the model due to biases that exist in the first half of the year. In agreement with MOPITT comparisons, the
model shows a large negative bias in winter and spring in the NH, with particularly large biases at stations located at higher
latitudes. This has been documented at Arctic surface sites previously (Shindell et al., 2008; Monks et al., 2015b). At latitudes
$>25°N$ the model has a normalised mean bias (NMB) of between -17.0 % and -38.1 % for RUN_2000 and -19.2 % and -33.1
% for RUN_2008. The winter/spring model bias is smaller at locations closer to the NH tropics resulting in lower NMB, with
the best overall model agreement found at Key Biscayne in Florida (NMB of 3.6% and-4.8 %). In the SH, the model has a
tendency to overestimate CO concentrations, particularly in the austral summer. Overall, the model has NMBs in the SH of
between 4.7 % and 37.3 % for RUN_2000 and -0.8 % and 30.3 %, with the best model performance at Easter Island in the



Pacific Ocean, and the worst at Cape Grim, Australia. In the SH, RUN_2008 has consistently lower NMBs, most likely due to lower emissions.

The 26 multi-model study by Shindell et al. (2006) found that models have a negative bias of between 20–80 ppbv at Alert in the Arctic during winter/spring and a more persistent positive bias throughout the year of up to 20–25 ppbv at Cape Grim,
exhibiting a similar transition from a negative bias in the NH to a positive bias in the SH to that found in TOMCAT. TOMCAT is within the bias range at Alert, with a winter negative bias of up to ~50 ppbv, and at the upper end of the bias range at Cape Grim with up to ~25 ppbv at Cape Grim. This and the MOPITT comparisons shows that these model biases exist at the surface and throughout the free troposphere, and are generally consistent with biases found in other chemical transport models.

### 4.4  Evaluaton of ozone

Ozonesonde data is compared to simulated $O_3$ in Figure 6. The data has been separated into three different altitude and latitude bands. The model overestimates $O_3$ at higher NH and SH latitudes in the highest altitude band (NMB of 22 % to 43.2 %), possibly due to too much downward mixing of stratospheric $O_3$ in the model at these altitudes. TOMCAT also overestimates $O_3$ at the surface in the tropics (NMB of 14.4 % to 16.7 %), but the model lies within the range of observations. Elsewhere, the model has a negative bias (NMB of -1.2 % to -24.6%), but lies within the range of observations at several times of the
25 year. Most of the negative bias in the higher latitudes is being driven by wintertime underestimates in $O_3$ in both the SH and NH. Young et al. (2013) compared the multi-model ACCMIP mean $O_3$ to ozonesonde data and found that it is also negatively biased in the SH during the winter months, but overestimated $O_3$ in the NH high latitudes during winter.

This low TOMCAT bias in wintertime $O_3$ can also be seen in surface data located at high latitudes (see Figure 7). TOMCAT has a negative $O_3$ bias of ~10–15 ppbv during the NH boreal winter at Heimaey, Iceland and during the SH austral winter at
30 Arrival Heights and South Pole (NMB of -19.7 % to -26.3%). All three of these stations are located near the poles suggesting that the model may have difficulties reproducing $O_3$ photochemistry in the winter in remote, dark and cold regions or the model may deposit too much $O_3$ onto snow/ice covered surfaces. Whilst most models in the POLMIP study were also negatively biased at the Summit observatory in the Arctic during winter, TOMCAT simulated some of the lowest concentrations (Monks et al., 2015b). Outside of the poles, the model simulates concentrations of $O_3$ that agree well with observations at this time of the year (see Figure 7), with the worst agreement occurring in the NH summer, where the model tends to overestimate concentrations. This is a common feature in models in the NH during summer, which has been identified to be paticularly pronounced over Eastern U.S. (e.g., Ellingsen et al., 2008; Fiore et al., 2009; Yu et al., 2010). $O_3$ at the surface is also compared at 44 EPA CASTNET stations located in the U.S. (Figure 8). This high bias over the U.S. is clearly evident, with a large mean
bias (MB) of 26.8/32.3 ppbv (NMB of 89.9/107.6 %). The best agreement is seen in winter (MB<-4.6 ppbv, NMB<-14.6 %). ValMartin et al. (2014) showed that model summertime $O_3$ biases could be reduced from 44 % to 28 % over the U.S. and from 25 % to 14 % over Europe when improvements were made to the coupled land-atmosphere model's deposition scheme. This suggests that using a more sophisticated deposition scheme coupled to a land model may improve TOMCAT simulations of summertime $O_3$.



RUN_2008 sub-column $O_3$ between 0–6 km (up to ∼500 hPa) is compared to GOME-2 retrievals in Figure 9. MB errors that are greater than the satellite error are highlighted with green polygons. In DJF, GOME-2 measures the highest concentrations of $O_3$ (∼25 DU) in regions near $O_3$ precursor emissions and enough sunlight to initiate photochemistry at this time of year (e.g. India, China and northern Africa; Figure 9b). TOMCAT shows negative MBs of up to -10 DU in several regions, with some of the larger biases being co-located with high observed $O_3$ concentrations (see Figure 9d). Comparisons to ozonesondes (see Figure 6) further support this and show that the model $O_3$ may be biased low (by 5–10 ppbv) in the tropical region at this time of year at altitudes between 750 hPa and 450 hPa (although the model does lie within the ozonesonde observed ranges). In JJA, the model bias is much smaller with very few significant MBs being highlighted (see Figure 9c). There is evidence that the model overestimates $O_3$ at this time of year over South-East U.S., in agreement with the CASTNET model-observation comparisons, and some evidence that $O_3$ is also overestimated near Cape Verde off the coast of Africa, as seen in Figure 7.

## 4.5 Evalulation of VOCs

Global maps of simulated propene ($C_3H_6$, Figure 10), ethene ($C_2H_4$, Figure 11), propane ($C_3H_8$, Figure 12) and ethane ($C_2H_6$, Figure 13) are shown averaged over 2–4 km and 5–8 km for both model runs. The lifetimes of these VOCs vary from a few hours to a few months, with propene having the shortest lifetime, followed by ethene, propane and then ethane (Rudolph et al., 1989). Due to the shorter lifetime of alkenes ($C_2H_4$, $C_3H_6$), the concentrations are mostly elevated near source regions due to limited long-range transport. When the lifetimes of the alkenes are longer due to lower OH (e.g. in winter), higher concentrations can be seen further from sources. In contrast, the longer lifetime of alkanes results in much more transport throughout the globe, with the highest concentrations being located in the NH due to higher anthropogenic emissions.

Overlaid on the maps are mean observed concentrations from several aircraft campaigns collected between 1992 and 2001 (Emmons et al., 2000). The spatial distribution of high and low concentrations of the alkanes and alkenes seems to be well represented by the model. However, it is noticeable that simulated concentrations, particularly of alkanes, are consistently lower than observed, suggesting a negative bias in the model. As these VOCs are emitted and not formed, this indicates that emissions are probably too low. A negative model bias in ethane and propane at several surface sites located in the NH has been shown previously to exist in several models (Emmons et al., 2015).

In Figure 14, measurements of ethene, ethane, propene, propane, toluene and butane are compared to simulated concentrations at the mountain site Hohenpeissenberg, Germany. The observations show a distinct seasonal cycle with peak concentrations in winter and spring, when OH concentrations are lower and the lifetimes of VOCs are longer, and a minimum in summer. For ethane and propane, the model captures the seasonal cycle (r=0.88–0.98), but shows a much smaller amplitude due to large negative biases, particularly in winter (NMB of -29.2 % to -80.5 %). For ethene and propene, the seasonal cycle is not well captured by the model due to enhancements in summer (r=0.13–0.70). The model simulates summer minimum concentrations in JJA elsewhere in the NH (see Figure 10–11). Therefore, this is likely due to incorrect local emissions or difficulties capturing local turbulent transport at this site, which is a common problem in models (Zhang et al., 2008; Feng et al., 2011). Similar to ethane and propane, the model also shows negative biases that are particularly large in winter. For toluene and butane, the model captures the seasonal cycle well (r=0.90–0.95), but some large biases are found (toluene: NMB



of 25.3/302 %, butane: NMB of 91/155.3%). For all the VOCs, the two simulations shows large differences in concentrations and biases, with RUN_2000 generally simulating lower VOC concentrations for all the the VOCs at this measurement site. This results in lower NMBs in toluene and butane in RUN_2000 but higher NMBs for the C2-C3 alkanes/alkenes. This is likely due to the fact that RUN_2000 and RUN_2008 emit different magnitudes of anthropogenic VOCs (see Table 1). In addition to this, RUN_2000 has larger biogenic C2-C3 alkene emissions, resulting in higher concentrations compared to RUN_2008 near biogenic emissions sources (e.g. over the Amazon) (see Figures 10–13). This highlights that regional model performance for VOCs is very dependent on the emission dataset chosen.

### 4.6  Evaluation of reactive nitrogen

Oxides of nitrogen (NOy) are important atmospheric pollutants and are key in the production of $O_3$. In addition, speciation of NOy is dependent on oxidative capacity, organic chemistry and heterogeneous chemistry. Hence, evaluation of speciated NOy is a valuable test of several inter-related aspects of model chemistry. Here we use observations of nitrogen oxide (NO), $NO_2$, nitric acid ($HNO_3$) and peroxyacetyl nitrate (PAN) to evaluate the model NOy.

In Figure 15 measurements of $NO_x$ (NO + $NO_2$) are compared to simulated concentrations at several European locations at a range of altitudes (see Figure 1c for locations). The observations generally show a minimum in summer and a maximum in winter, with the model showing variable skill in capturing the seasonal cycle depending on the surface station (r=0.08–0.89). Both TOMCAT simulations underestimate $NO_x$ concentrations (NMB of -30 % and -101.1 %), with RUN_2000 having the lowest overall bias due to larger concentrations. The largest biases occur in winter, resulting in a smaller seasonal cycle amplitude. As $NO_x$ is very short-lived, it is difficult for global models to reproduce observations due to coarse horizontal resolutions (Huijnen et al., 2010), which may partly explain the negative biases seen in the model.

Figure 16 shows DJF and JJA satellite OMI $NO_2$ column for 2008 and the TOMCAT MB from RUN_2008. Due to the short lifetime of $NO_2$, high concentrations are located near emission regions. In the NH, high concentrations are seen over Asia, North America and Europe, near some of the largest anthropogenic emission sources. In both seasons the model simulates concentrations that are too high over parts of Europe. This is in contrast to the surface comparisons shown in Figure 15, where the model underestimated $NO_x$ at several locations at altitudes below 3 km. However, most of these sites were located further west of the region where OMI shows a positive bias in the model. Large negative biases in $NO_2$ near China are seen in the model in the NH winter. This has been seen in several models previously when comparing to OMI and is thought to be due to anthropogenic emissions that are too low (Emmons et al., 2015). In contrast, TOMCAT has a positive model bias in this region during summer, most likely due to fire emissions that are too high, which has also been seen in multiple models being compared to OMI (Emmons et al., 2015).

In the SH, OMI observes the largest concentrations over the high biomass burning regions of South America, Africa and Australia. In these regions the model shows $NO_2$ concentrations that are too low during both seasons, suggesting emissions from fires are too low in the SH. This is in contrast to CO satellite comparisons, which suggested fire emissions are too high in this region (see Section 4.3). This therefore indicates that emission factors used to calculate fire emissions need to be further evaluated in the tropics and the SH.





PAN is formed from the oxidation of VOCs and reaction of the peroxyacetyl radical with $NO_2$. It is an important $NO_x$ reservoir and acts to transport and supply $NO_x$ to regions remote from emission sources, where it can contribute to $O_3$ production (e.g., Hudman et al., 2004). Figure 17 shows TOMCAT PAN peaks in the lower troposphere in spring and peaks in the upper troposphere in summer, which is in agreement with the GEOS-Chem model (Fischer et al., 2014) and upper tropospheric satellite observations from MIPAS-E (Moore and Remedios, 2010). The aircraft climatology shows that TOMCAT captures

the spatial variability of PAN well (see Figure 17). However, TOMCAT may overestimate concentrations of PAN over North America in winter. Pope et al. (2016) also found that TOMCAT PAN may be too high against aircraft data in the Arctic and over North America in winter. They also found that TOMCAT PAN overestimated upper tropospheric MIPAS satellite PAN in winter and spring in parts of the NH. A high spring bias in PAN has also been shown to be present in the GEOS-Chem CTM when compared to aircraft data (Fischer et al., 2014).

Figure 18 shows surface PAN comparisons at two locations in Europe at two different altitudes, with concentrations peaking in the spring. Both model simulations show a summer maximum and a winter minimum (r=0.8–0.85), with a ∼2 month lag in the peak concentrations. At Schauinsland (1205 m), both of the model simulations lie within the range of observations. At Zugspitze, which has lower observed concentrations, both model simulations fall outside of the range of observations during certain times of the year. RUN_2008 simulates the highest concentrations and as a result, overestimates PAN in summer at

both stations (NMB=43–77.5%). RUN_2000 has much better overall agreement with observations due to lower simulated concentrations (NMB=-4.0–17.4%). RUN_2000 also showed better agreement against the aircraft climatology in the NH due to lower concentrations. As already mentioned, the VOC emissions differ between the runs, which will affect PAN production. Differences in model meteorology between the years may also play a role.

$HNO_3$ is compared to the aircraft climatology in Figure 19. The spatial variability in $HNO_3$ is generally captured in the model

where observations exist. However, both RUN_2000 and RUN_2008 simulate $HNO_3$ concentrations that are higher than those from the aircraft climatology over parts of North America during DJF and MAM. In an Arctic model intercomparison project (POLMIP), TOMCAT had some of the highest concentrations of PAN and $HNO_3$ compared to other models (Emmons et al., 2015), suggesting that TOMCAT $NO_y$ production may be more efficient than other models, and/or loss may be underestimated in TOMCAT in the mid to high latitude NH regions. This is important for $O_3$ production and warrants further investigation.

## 5   Summary

The TOMCAT chemical transport model has been updated with the Extended Tropospheric Chemistry scheme, adding the degradation of ethene, propene, toluene and butane. Monoterpene chemistry has also been added based on MOZART-3 chem-

5   istry. Two model simulations, which differ in their boundary conditions but both use the new chemistry scheme, are used to evaluate the model against a range of surface, satellite, aircraft and balloon measurements. The model is generally able to capture the main spatial and seasonal features of high and low concentrations of CO, $O_3$, VOCs and reactive nitrogen. However, several negative and positive biases are present in TOMCAT during certain times of the year and at certain locations. Some of





these biases are prevalent in current state-of-the-art chemistry models, but some biases that are specific to TOMCAT are also highlighted.

TOMCAT global mean tropospheric OH ($1.07$-$1.08 \times 10^6$ molecules/cm$^3$) is higher than estimates inferred from MCF observations ($0.94$-$1.0 \times 10^6$ molecules/cm$^3$). However, this is a common feature across chemistry models and the TOMCAT global mean OH is at the lower end of concentrations reported in previous multi-model intercomparison projects ($1.08$-$1.17 \times 10^6$ molecules/cm$^3$). TOMCAT has the highest concentrations (in molec./cm$^3$) of OH in the lower tropical troposphere, which is in contrast to the ACCMIP multi-model mean OH, which has the highest OH concentrations in the tropical upper troposphere. However, observationally-constrained OH shows the highest concentration of OH in the middle tropical troposphere. Further to this, TOMCAT has a higher NH/SH OH ratio (1.37) compared to the ratio inferred from MCF observations (0.98), which is again a common feature in chemistry models, with TOMCAT being at the upper limit of the multi-model mean value calculated from the ACCMIP models ($1.28 \pm 0.1$). This suggests that simulated OH in current chemistry is largely uncertain.

TOMCAT CO is negatively biased during winter and spring in the NH when compared to MOPITT and surface observations. In contrast, CO is positively biased throughout the year in the SH. The negative bias in the NH is a common feature in chemistry models and TOMCAT lies well within the range of biases found in other models. The TOMCAT SH positive bias is at the upper range of positive biases reported in other models, with some models reporting negative biases. Underestimated emissions in the NH are thought to play a role in the negative NH CO bias, whilst comparisons with MOPITT suggest that TOMCAT fire emissions may be too high in the SH. Biases could also be reduced by lowering the NH/SH OH ratio, where a lower OH concentration in the NH will increase the lifetime of CO and/or a higher OH concentration in the SH would decrease the lifetime of CO.

TOMCAT is able to capture the seasonality of O$_3$ in most locations, with the model lying within the range of observations made during balloon soundings during most times of the year. The notable exceptions to this are: 1) at high latitudes during winter conditions, where TOMCAT simulates O$_3$ that is negatively biased by up to 15 ppbv when compared to both surface and ozonesonde measurements and 2) in the NH during summer, where TOMCAT is positively biased by up to 32 ppbv over North America when compared to surface sites. GOME-2 satellite data shows that model performance is better in JJA compared to DJA, where the model underestimates O$_3$ by up to 10 DU in regions with high observed O$_3$ concentrations near Asia and Africa.

Comparison of simulated VOCs, NO$_x$, PAN and HNO$_3$ to observations highlights the sensitivity of model performance to model boundary conditions and emissions, with a large range in concentrations between RUN_2000 and RUN_2008. Comparison with an aircraft climatology shows that the model generally captures the spatial and temporal variability of these species. However some biases are found. VOC aircraft and surface measurements show large negative biases in simulated winter/spring C2–C3 alkanes and alkenes, which is likely driven by underestimated anthropogenic emissions. This has been seen previously for ethane and propane in several models in the NH. In contrast, RUN_2000 showed very good agreement with the seasonality and magnitude of surface measurements of toluene and butane in Europe.

TOMCAT is also able to capture the seasonal minima and maxima of PAN that vary with altitude. However, TOMCAT has a 2 month lag in peak PAN concentrations when compared to mountain site observations made in Europe. The aircraft climatology





also shows that wintertime PAN may be overestimated in winter and $HNO_3$ may be overestimated in winter and spring in some regions over North America. Whilst the aircraft climatology is useful for general comparisons it is noted that the model simulations do not match the observations in time and therefore some discrepancies are expected. For $NO_x$, the amplitude of

15 simulated $NO_2$ at European sites is much smaller than observed due to large negative model biases, particularly in winter. This is likely to be at least partly due to the very short lifetime of $NO_x$ and the coarse model grid. Satellite OMI $NO_2$ showed regional differences in TOMCAT biases, with negative biases existing over China in DJF (possibly due to anthropogenic emissions) and South America and Africa (possibly due to fire emissions), and positive biases over Europe in DJF and JJA. The biases over Asia have been shown to exist in several other models when using the same emissions as used for RUN_2008. Further to this, models have been shown previously to vary widely in the simulation of species such as $HNO_3$, PAN and acetaldehyde.

Therefore, observations of such species that are collected continuously throughout the year at several locations globally would be valuable in evaluating chemical transport models in the future and understanding model biases in $O_3$.

## 6 Code Availability

TOMCAT/SLIMCAT (www.see.leeds.ac.uk/tomcat) is a UK community model. It is available to UK (or NERC-funded) researchers who normally access the model on common facilities or who are helped to install it on their local machines. As it

is a complex research tool, new users will need help to use the model optimally. We do not have the resources to release and support the model in an open way. Any potential user interested in the model should contact Martyn Chipperfield. The model updates described in this paper are included in the standard model library.

*Acknowledgements.* Funding for this work was provided by the NERC EurEx project (NE/H020241/1) and TOMCAT model simulations were performed on the UK Archer HPC system. We would like to thank the many providers of observational data that has been used in this

paper. Specifically, the World Ozone and Ultraviolet Data Center (WOUDC), the National Oceanic and Atmospheric Administration Earth System Research Laboratory, Global Monitoring Division (NOAA ESRL, GMD) and the Southern Hemisphere ADditional OZonesondes (SHADOZ) for data that was used in the ozonesonde climatology. The MOPITT team that provided CO satellite data and L.K. Emmons for providing code to process the MOPITT data. The OMI satellite group for use of tropospheric $NO_2$ column data. The United States Environmental Protection Agency (EPA) Clean Air Status and Trends Network (CASTNET) for $O_3$ surface measurements. The Global Atmosphere

Watch (GAW) program for the use of the World Data Centre for Greenhouse Gases (WDCGG), which provided surface measurements of CO, $O_3$, $NO_x$, PAN and VOCs. WDCGG providers whose data was used includes NOAA/ESRL, University of York, UBA, Empa, RIVM and DWD.



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





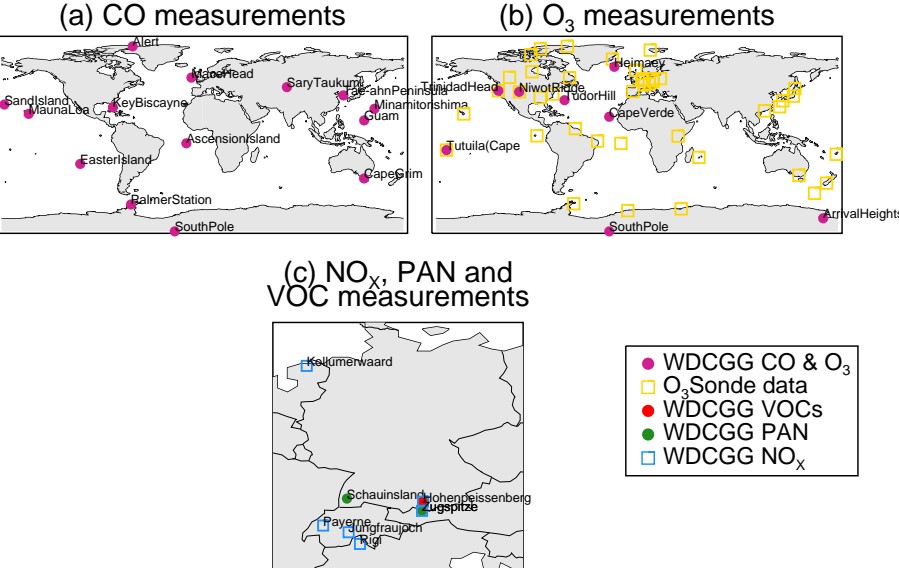

**Figure 1.** Location of WDCGG surface observatories and ozonesonde release sites used to evaluate the model for a) CO, b) $O_3$, and c) PAN, $NO_x$ and VOCs.

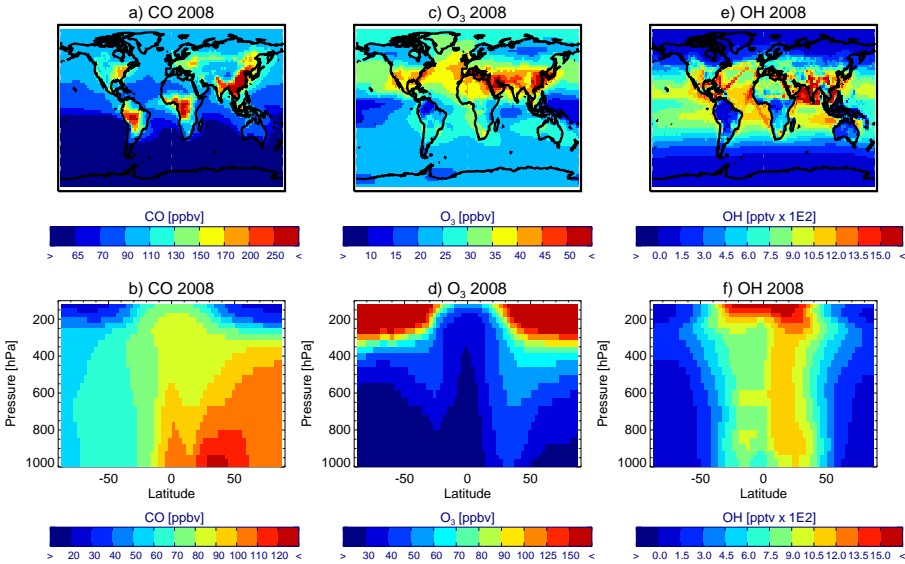

**Figure 2.** Concentrations of annual surface mean and annual zonal mean CO (a,b), $O_3$ (c,d) and OH (e,f) from TOMCAT simulation RUN_2008.



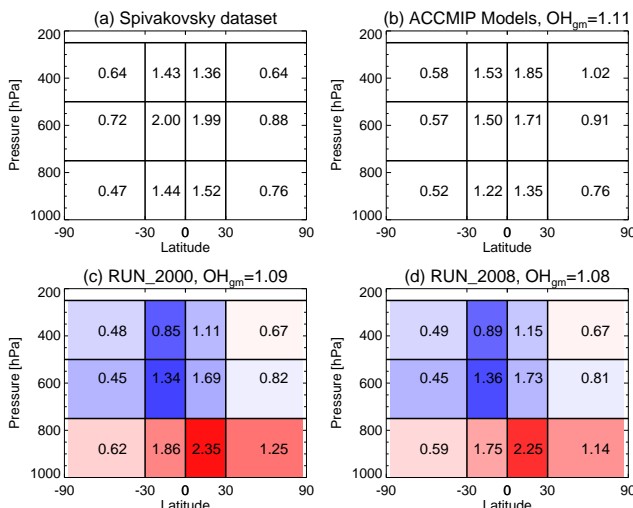

**Figure 3.** Regional mean OH concentrations ($\times 10^6$ molecules/cm$^3$) split into subsections as recommended by Lawrence et al. (2001). a) OH estimated from methyl chloroform observations from Spivakovsky et al. (2000), b) ACCMIP multi-model mean simulated OH concentrations from Naik et al. (2013) and TOMCAT-simulated OH concentrations for c) RUN_2000 and d) RUN_2008. The air-mass-weighted global mean tropospheric OH (OH$_g$m) is indicated above each plot for panels b)-d). In TOMCAT, the troposphere was defined as the area below a climatological tropopause ($p = 300 - 215(cos(lat))^2$) (as discussed in Lawrence et al. (2001)) and for ACCMIP it was defined as below 200 hPa. The colours in c) and d) are scaled according to the difference from a) with the darkest blue representing the largest negative differences and the darkest red representing the largest positive differences.




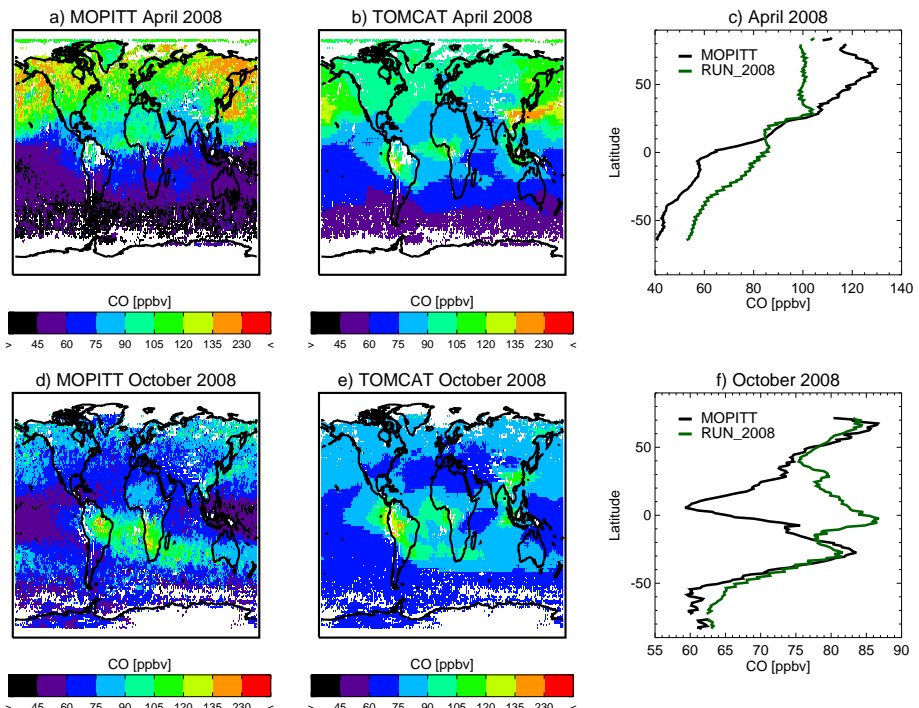

**Figure 4.** 500 hPa CO concentrations (ppbv) observed by MOPITT (a,c) and simulated by TOMCAT (RUN_2008) (b,d) during April and October 2008. The zonal mean concentrations at 500 hPa are also shown (e,f; data only shown when there is 25% coverage in a given latitude band). MOPITT Averaging kernels have been applied to the TOMCAT fields.





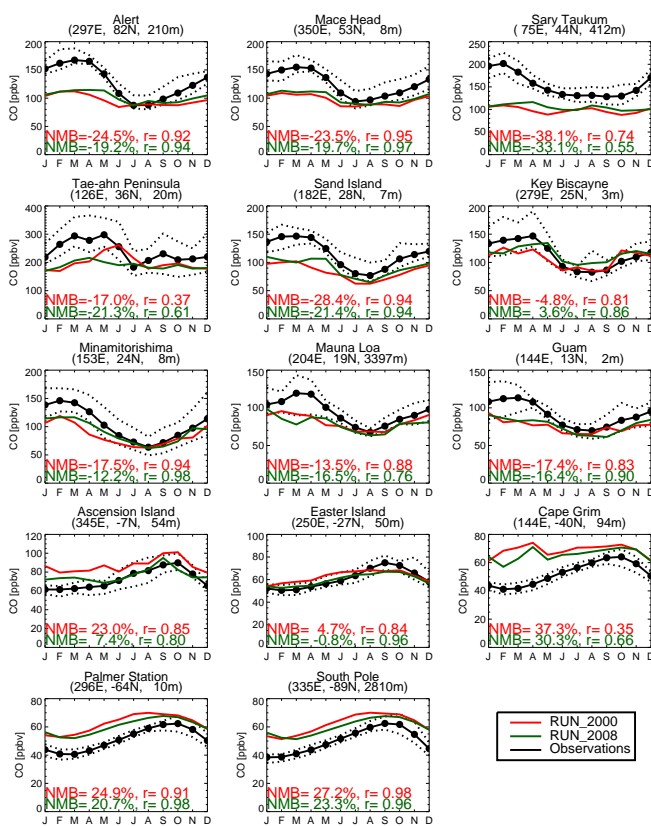

**Figure 5.** Observed and simulated CO (ppbv) at a range of WDCGG surface observatories located throughout the globe. The observations are shown as an average (black solid line) and as minimum and maximum concentrations (black dashed lines) of all available data between 2000 and 2008. The sites are arranged by latitude from north to south.





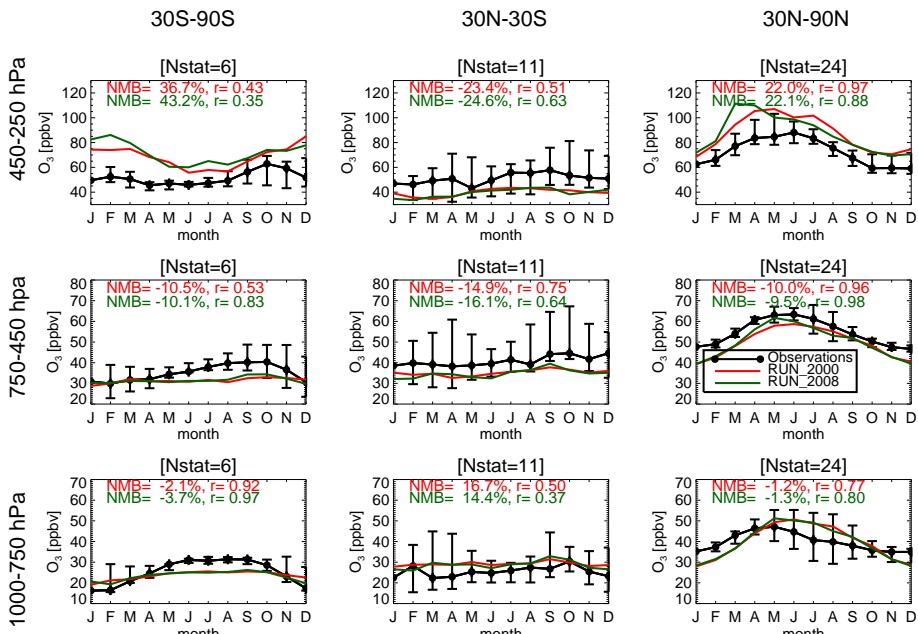

**Figure 6.** Median O$_3$ concentrations (ppbv) taken from the Tilmes et al. (2012) ozonesonde climatology compared to concentrations from RUN_2000 and RUN_2008. The data is average over three latitude ranges (left to right) and three pressure level ranges (top to bottom), where the error bars show the 25th and 75th percentiles of the observed concentrations and Nstat gives the number of sonde release sites located within each latitude range.





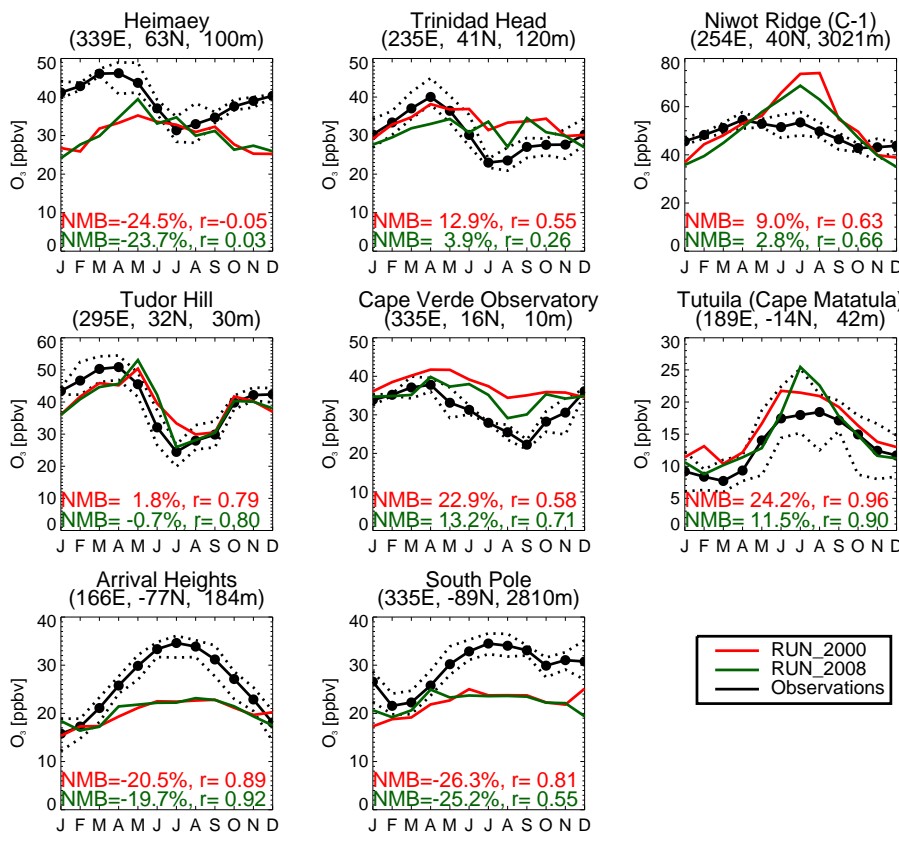

**Figure 7.** Observed and simulated $O_3$ (ppbv) at several WDCGG surface observatories. The observations are shown as an average (black solid line) and as minimum and maximum concentrations (black dashed lines) of all available data between 2000 and 2008. The sites are arranged by latitude from north to south.

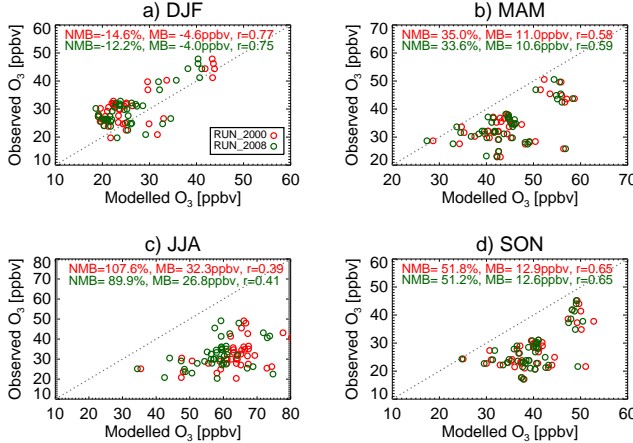

**Figure 8.** Scatter plots of seasonal mean observed and simulated $O_3$ concentrations (ppbv) at CASTNET EPA monitoring stations located in North America. The observations are a mean of 2000 to 2008 available data.





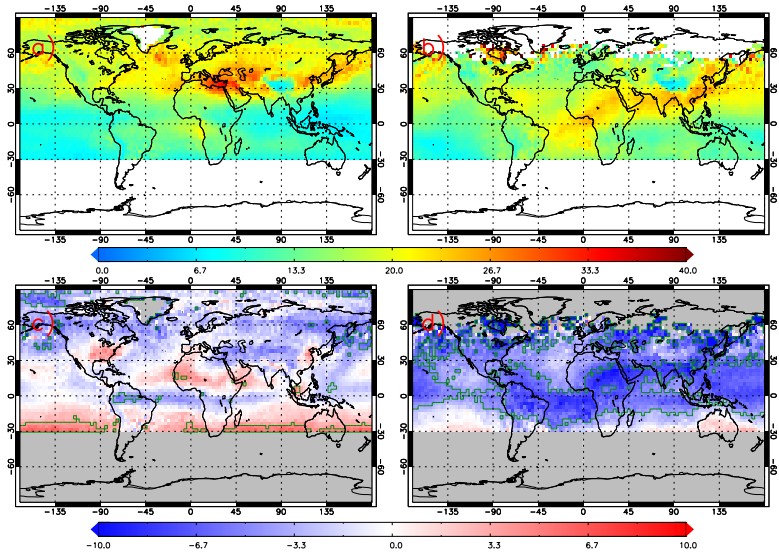

**Figure 9.** GOME-2 sub-column $O_3$ (0-6 km, DU) on the TOMCAT $2.8° \times 2.8°$ grid for a) June-July-August 2008 (JJA) and b) December-January-February 2008 (DJF). c) and d) show the difference in concentrations between TOMCAT RUN_2008 and GOME-2. The green polygons are where the mean bias (MB) is greater than the satellite error.

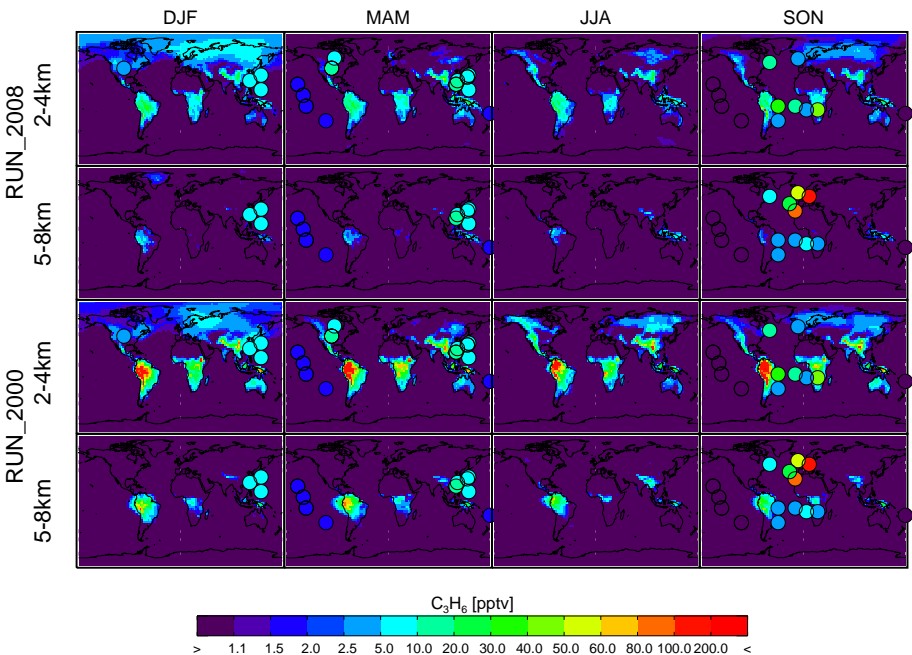

**Figure 10.** Maps of simulated seasonal mean concentrations of propene ($C_3H_6$; pptv) averaged over two altitude bands (2–4 km and 5–8 km), with overlaid circles coloured according to concentrations from the aircraft climatology of Emmons et al. (2000).




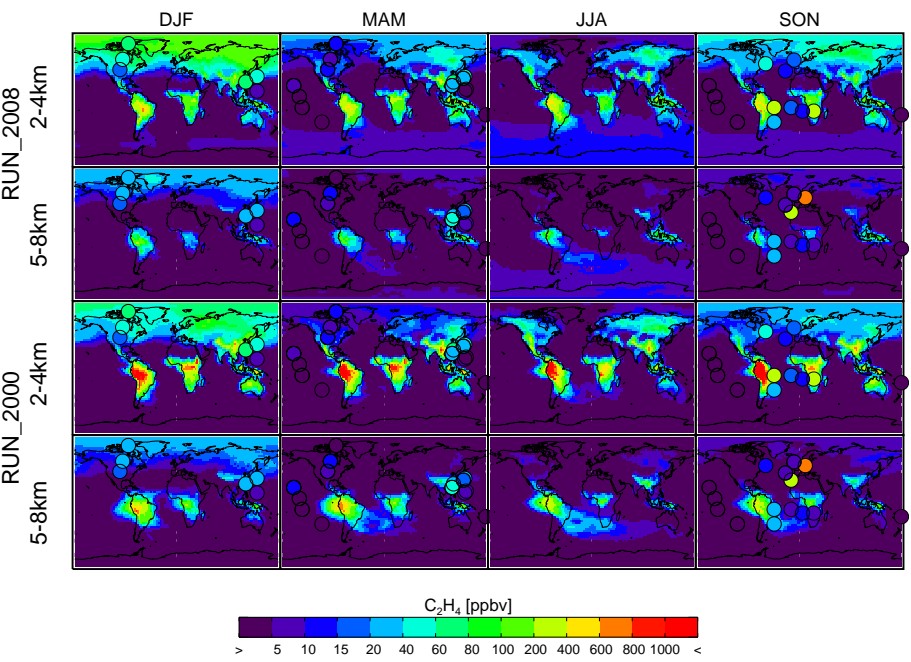

**Figure 11.** As Figure 10, but for ethene ($C_2H_4$; pptv).

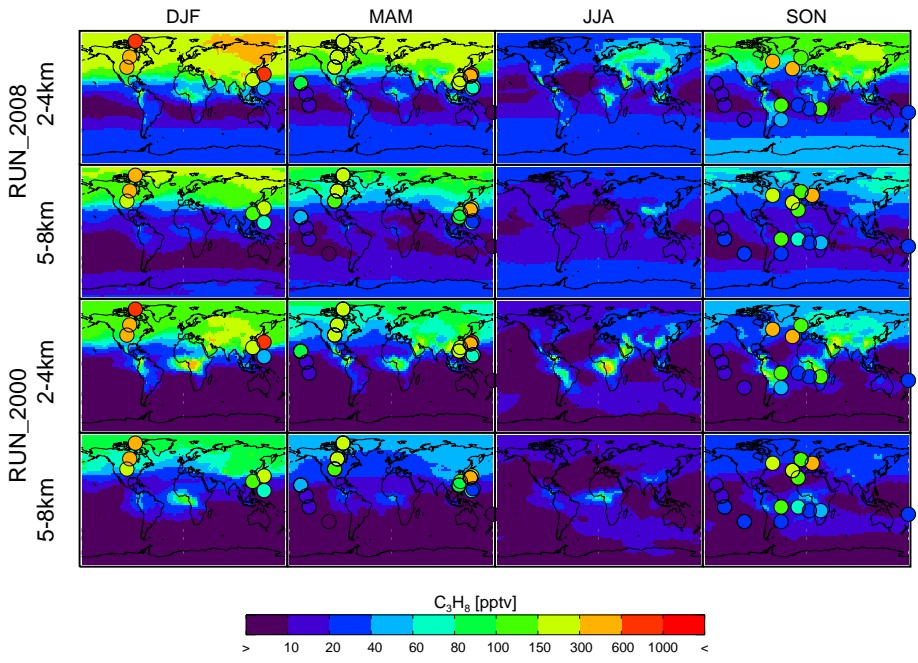

**Figure 12.** As Figure 10, but for propane ($C_3H_8$; pptv).





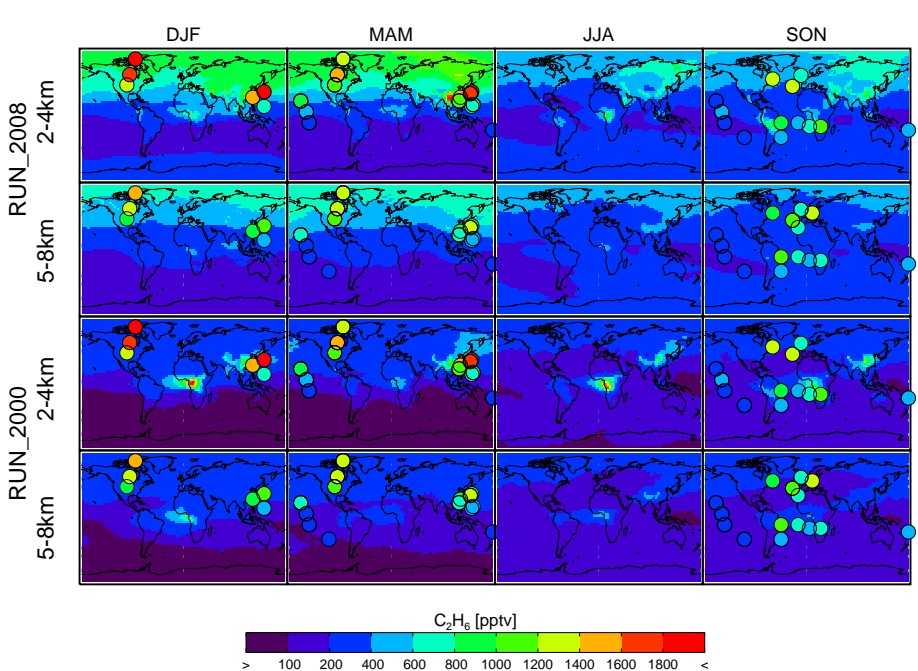

**Figure 13.** As Figure 10, but for ethane ($C_2H_6$; pptv).





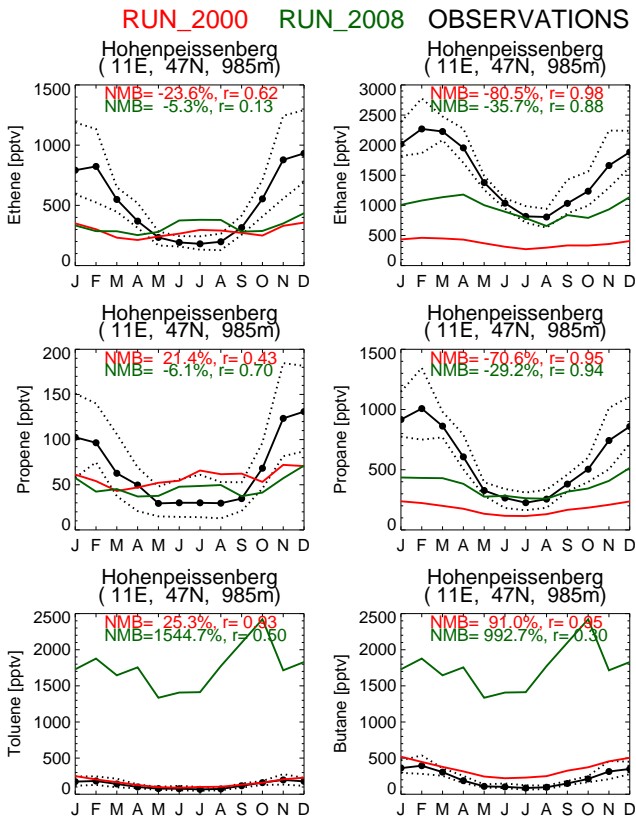

**Figure 14.** Observed and simulated VOCs (pptv) at the European high altitude observatory, Hohenpeissenberg. The observations are shown as an average (black solid line) and as minimum and maximum concentrations (black dashed lines) of all available data between 2000 and 2008.





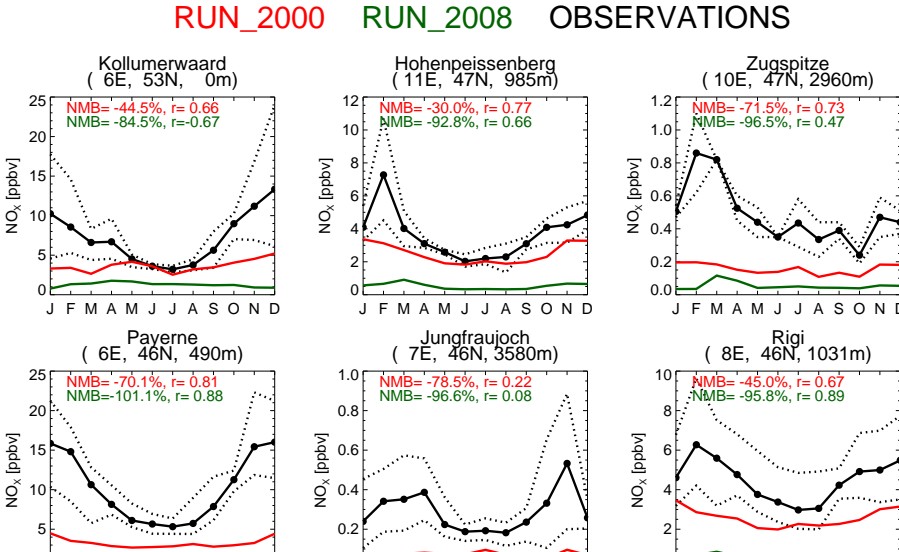

**Figure 15.** Observed and simulated NO$_x$ (ppbv) at several European surface observatories. The observations are shown as an average (black solid line) and as minimum and maximum concentrations (black dashed lines) of all available data between 2000 and 2008.

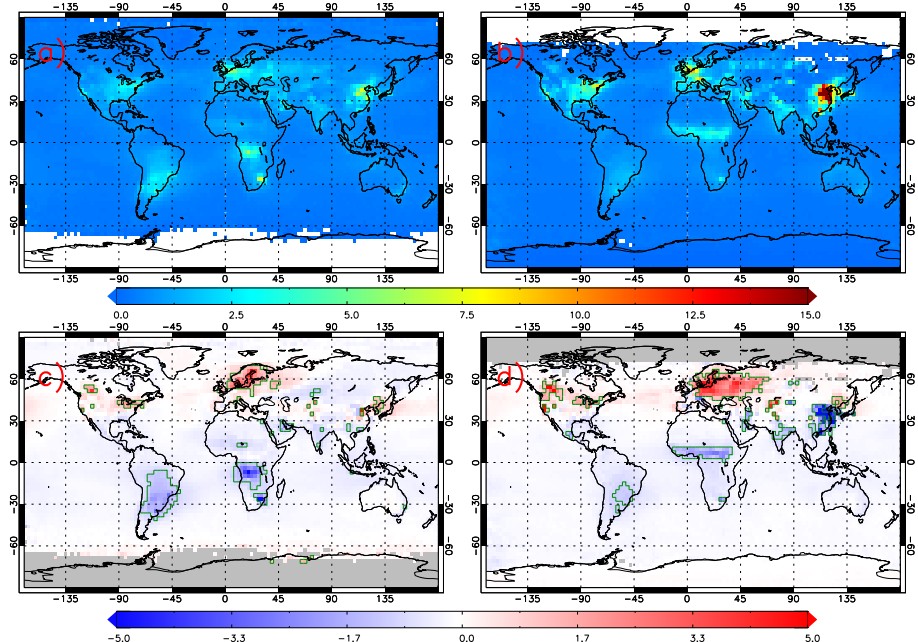

**Figure 16.** OMI tropospheric column NO$_2$ (x10$^{15}$ molecules cm$^{-2}$) on the TOMCAT 2.8° × 2.8° grid for a) June-July-August (2008) and b) January-February-December (2008), c) and d) are the TOMCAT OMI tropospheric column NO$_2$ mean bias (MB) for the same periods. The green polygons are where the |MB| > satellite error.





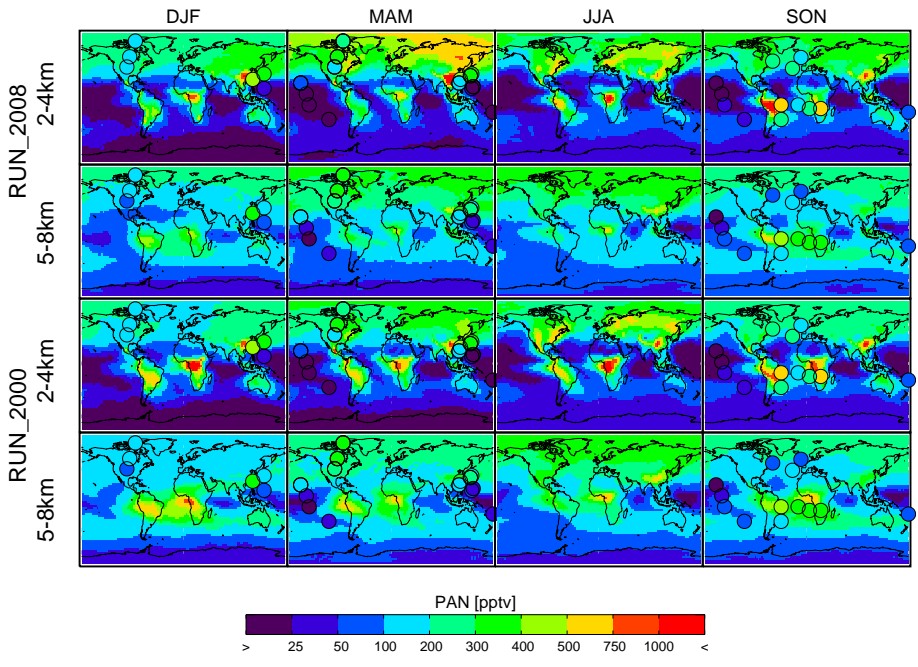

**Figure 17.** Maps of simulated seasonal mean concentrations of peroxyacetyl nitrate (PAN; pptv) averaged over two altitude bands (2–4 km and 5–8 km), with overlaid circles coloured according to concentrations from the aircraft climatology of Emmons et al. (2000).

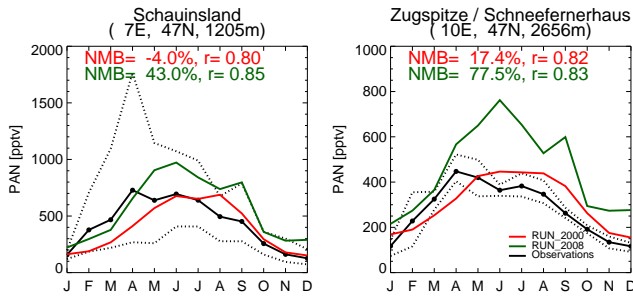

**Figure 18.** Observed and simulated PAN (pptv) at two European high altitude observatories. The observations are shown as averages (black solid line) and as minimum and maximum concentrations (black dashed lines) of all available data between 2000 and 2008.





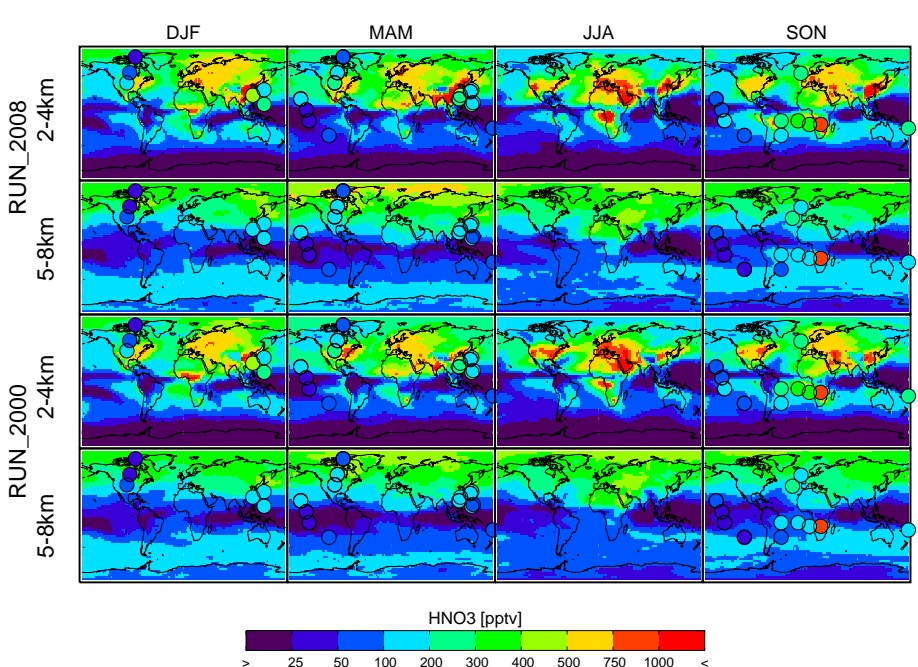

**Figure 19.** As Figure 17, but for nitric acid (HNO$_3$; pptv).



**Table 1.** TOMCAT annual global emissions (Tg(species)/year).

| Species | RUN_2000 | | | | | | RUN_2008 | | | | | |
|---|---|---|---|---|---|---|---|---|---|---|---|---|
| | Anthropogenic | Fires | Biogenic | Ocean | Soil | Total | Anthropogenic | Fires | Biogenic | Ocean | Soil | Total |
| CO | 609.27 | 266.22 | 84.25 | 19.87 | | 979.61 | 595.27 | 331.62 | 76.57 | 20.01 | | 1023.47 |
| Ethene | 7.74 | 3.80 | 29.33 | | | 40.87 | 6.81 | 2.84 | 16.70 | 1.40 | | 27.75 |
| Ethane | 3.34 | 1.94 | 0.34 | | | 5.62 | 6.34 | 1.67 | 0.14 | 0.98 | | 9.14 |
| Propene | 4.47 | 2.09 | 16.33 | | | 22.90 | 3.04 | 1.57 | 6.10 | 1.52 | | 12.23 |
| Propane | 4.04 | 1.03 | 0.03 | | | 5.11 | 5.68 | 0.38 | 0.02 | 1.30 | | 7.37 |
| Toluene | 7.05 | 2.44 | 0.26 | | | 9.75 | 25.34 | 10.66 | 0.26 | | | 36.26 |
| Butane | 10.41 | 0.61 | | | | 11.02 | 12.38 | 0.60 | | | | 12.98 |
| Formaldehyde | 3.18 | 3.77 | 5.10 | | | 12.05 | 2.99 | 4.13 | 4.03 | | | 11.15 |
| Acetone | 2.21 | 1.85 | 45.11 | | | 49.16 | 0.54 | 1.86 | 28.58 | | | 30.98 |
| Acetaldehyde | 1.92 | 2.58 | 21.27 | | | 25.77 | 2.00 | 4.55 | 11.20 | | | 17.75 |
| Methanol | 5.71 | 6.75 | 103.47 | | | 115.93 | 0.93 | 5.38 | 159.87 | | | 166.18 |
| Isoprene | | 0.38 | 543.70 | | | 544.08 | | 0.80 | 525.84 | | | 526.64 |
| Monoterpenes | | 0.27 | 158.84 | | | 159.11 | | 0.28 | 97.10 | | | 97.37 |
| NO$_x$ | 104.87 | 8.15 | | | 26.29 | 139.31 | 107.73 | 19.41 | | | 16.31 | 143.46 |



Table 2: Chemical species treated in the tropospheric chemistry scheme of the TOMCAT CTM. If the species are emitted, dry deposited or wet deposited a Y is in the relevant column. The family column indicates which short-lived species are grouped together for advection and chemistry. TOMCAT abbreviations: Me=$CH_3$, Et=$C_2H_5$, Pr=$C_3H_7$, MACR=lumped species(methacrolein, methyl vinyl ketone and other C4 carbonyls), HACET=hydroxyacetone, MGLY=methylglyoxal, NALD=nitrooxy acetaldehyde, TERP=generic terpene compound, AROM=generic aromatic compound, MEK=Methyl ethyl ketone, Prpe=$C_3H_7O$, ONIT=organic nitrate, S=stratospheric tracer (TOMCAT species 39–43).

| | Species | Family | Dry Deposited? | Wet Deposited? | Emitted? |
|---|---|---|---|---|---|
| 1 | $O(^3P)$ | Ox | | | |
| 2 | $O(^1D)$ | Ox | | | |
| 3 | $O_3$ | Ox | Y | | |
| 4 | NO | NOx | Y | | |
| 5 | $NO_3$ | NOx | Y | Y | |
| 6 | $NO_2$ | NOx | Y | | Y |
| 7 | $N_2O_5$ | | Y | Y | |
| 8 | $HO_2NO_2$ | | Y | Y | |
| 9 | $HONO_2$ | | Y | Y | |
| 10 | OH | | | | |
| 11 | $HO_2$ | | | Y | |
| 12 | $H_2O_2$ | | Y | Y | |
| 13 | $CH_4$ | | | | Y |
| 14 | CO | | Y | | Y |
| 15 | HCHO | | Y | Y | Y |
| 16 | MeOO | | | Y | |
| 17 | $H_2O$ | | | | |
| 18 | MeOOH | | Y | Y | |
| 19 | HONO | | Y | Y | |
| 20 | $C_2H_6$ | | | | Y |
| 21 | EtOO | | | | |
| 22 | EtOOH | | Y | Y | |
| 23 | MeCHO | | Y | | Y |
| 24 | $MeCO_3$ | | | | |
| 25 | PAN | | Y | | |
| 26 | $C_3H_8$ | | | | Y |
| 27 | n-PrOO | | | | |
| 28 | i-PrOO | | | | |
| 29 | n-PrOOH | | Y | Y | |
| 30 | i-PrOOH | | Y | Y | |
| 31 | EtCHO | | Y | | |
| 32 | $EtCO_3$ | | | | |
| 33 | $Me_2CO$ | | Y | | Y |
| 34 | $MeCOCH_2OO$ | | | | |
| 35 | $MeCOCH_2OOH$ | | Y | Y | |
| 36 | PPAN | | Y | | |
| | Continued on next page | | | | |





**Table 2 – continued from previous page**

| | TOMCAT Species | Family | Dry Deposited? | Wet Deposited? | Emitted? |
|---|---|---|---|---|---|
| 37 | $MeONO_2$ | | | | |
| 38 | $O(^3P)S$ | Sx | | | |
| 39 | $O(^1D)S$ | Sx | | | |
| 40 | $O_3S$ | Sx | Y | | |
| 41 | NOXS | | Y | | |
| 42 | $HNO_3S$ | | Y | Y | |
| 43 | NOYS | | Y | Y | |
| 44 | $C_5H_8$ | | | | Y |
| 45 | $C_{10}H_{16}$ | | | | Y |
| 46 | TERPOOH | | Y | Y | |
| 47 | $ISO_2$ | | | | |
| 48 | ISOOH | | Y | Y | |
| 49 | ISON | | Y | Y | |
| 50 | MACR | | Y | | |
| 51 | $MACRO_2$ | | | | |
| 52 | MACROOH | | Y | Y | |
| 53 | MPAN | | Y | | |
| 54 | HACET | | Y | Y | |
| 55 | MGLY | | Y | Y | |
| 56 | NALD | | Y | | |
| 57 | HCOOH | | Y | Y | |
| 58 | $MeCO_3H$ | | Y | Y | |
| 59 | $MeCO_2H$ | | Y | Y | |
| 60 | MeOH | | Y | Y | Y |
| 61 | $TERPO_2$ | | | | |
| 62 | $C_2H_4$ | | | | Y |
| 63 | $C_2H_2$ | | | | Y |
| 64 | $C_4H_{10}$ | | | | Y |
| 65 | $C_3H_6$ | | | | Y |
| 66 | AROM | | | | Y |
| 67 | MEK | | | | |
| 68 | MeCOCOMe | | Y | Y | |
| 69 | BtOO | | | | |
| 70 | PrpeOO | | | | |
| 71 | $AROMO_2$ | | | | |
| 72 | MEKOO | | | | |
| 73 | BtOOH | | Y | Y | |
| 74 | PrpeOOH | | Y | Y | |
| 75 | AROMOOH | | Y | Y | |
| 76 | MEKOOH | | Y | Y | |





**Table 2 – continued from previous page**

|  | TOMCAT Species | Family | Dry Deposited? | Wet Deposited? | Emitted? |
|---|---|---|---|---|---|
| 77 | ONIT |  |  |  |  |
| 78 | EtCO$_3$H |  |  |  |  |
| 79 | EtCO$_2$H |  |  |  |  |





Table 3: TOMCAT gas-phase bimolecular reactions. $T$ is the model grid-box temperature in kelvins. Reaction rate references 1: Atkinson et al. (a), 2: Atkinson et al. (b), 3: Atkinson et al. (c), 4: MCM (2004), 5: Tyndall et al. (2001), 6: Ravishankara et al. (2002), 7: Pöschl et al. (2000), 9: Kinnison et al. (2007), 10: Folberth et al. (2006).

| | Reactants | Products | $k$ | Reference |
|---|---|---|---|---|
| 1 | $HO_2 + NO$ | $\rightarrow OH + NO_2$ | $3.60\times10^{-12}\exp(\frac{270}{T})$ | 2 |
| 2 | $HO_2 + NO_3$ | $\rightarrow OH + NO_2$ | $4.00\times10^{-12}$ | 2 |
| 3 | $HO_2 + O_3$ | $\rightarrow OH + O_2$ | $2.03\times10^{-16}(\frac{T}{300})^{4.57}\exp(\frac{693}{T})$ | 2 |
| 4 | $HO_2 + HO_2$ | $\rightarrow H_2O_2$ | $2.20\times10^{-13}\exp(\frac{600}{T})$ | 2 |
| 5 | $HO_2 + MeOO$ | $\rightarrow MeOOH$ | $3.80\times10^{-13}\exp(\frac{780}{T})$ | 2 |
| 6 | $HO_2 + MeOO$ | $\rightarrow HCHO$ | $3.80\times10^{-13}\exp(\frac{780}{T})$ | 2 |
| 7 | $HO_2 + EtOO$ | $\rightarrow EtOOH$ | $3.80\times10^{-13}\exp(\frac{900}{T})$ | 2 |
| 8 | $HO_2 + MeCO_3$ | $\rightarrow MeCO_3H$ | $2.08\times10^{-13}\exp(\frac{980}{T})$ | 2 |
| 9 | $HO_2 + MeCO_3$ | $\rightarrow MeCO_2H + O_3$ | $1.04\times10^{-13}\exp(\frac{980}{T})$ | 2 |
| 10 | $HO_2 + MeCO_3$ | $\rightarrow OH + MeOO$ | $2.08\times10^{-13}\exp(\frac{980}{T})$ | 2 |
| 11 | $HO_2 + n\text{-}PrOO$ | $\rightarrow n\text{-}PrOOH$ | $1.51\times10^{-13}\exp(\frac{1300}{T})$ | 4 |
| 12 | $HO_2 + i\text{-}PrOO$ | $\rightarrow i\text{-}PrOOH$ | $1.51\times10^{-13}\exp(\frac{1300}{T})$ | 4 |
| 13 | $HO_2 + EtCO_3$ | $\rightarrow O_2 + EtCO_3H$ | $3.05\times10^{-13}\exp(\frac{1040}{T})$ | 4 |
| 14 | $HO_2 + EtCO_3$ | $\rightarrow O_3 + EtCO_2H$ | $1.25\times10^{-13}\exp(\frac{1040}{T})$ | 4 |
| 15 | $HO_2 + MeCOCH_2OO$ | $\rightarrow MeCOCH_2OOH$ | $1.36\times10^{-13}\exp(\frac{1250}{T})$ | 4 |
| 16 | $MeOO + NO$ | $\rightarrow HO_2 + HCHO + NO_2$ | $2.95\times10^{-12}\exp(\frac{285}{T})$ | 2 |
| 17 | $MeOO + NO$ | $\rightarrow MeONO_2$ | $2.95\times10^{-15}\exp(\frac{285}{T})$ | 2 |
| 18 | $MeOO + NO_3$ | $\rightarrow HO_2 + HCHO + NO2$ | $1.30\times10^{-12}$ | 2 |
| 19 | $MeOO + MeOO$ | $\rightarrow MeOH + HCHO$ | $1.03\times10^{-13}\exp(\frac{365}{T})$ | 4 |
| 20 | $MeOO + MeOO$ | $\rightarrow HO_2 + HO_2 + HCHO + HCHO$ | $1.03\times10^{-13}\exp(\frac{365}{T})$ | 2 |
| 21 | $MeOO + MeCO_3$ | $\rightarrow HO_2 + HCHO + MeOO$ | $1.80\times10^{-12}\exp(\frac{500}{T})$ | 2 |
| 22 | $MeOO + MeCO_3$ | $\rightarrow MeCO_2H + HCHO$ | $2.00\times10^{-13}\exp(\frac{500}{T})$ | 2 |
| 23 | $EtOO + NO$ | $\rightarrow MeCHO + HO_2 + NO_2$ | $2.60\times10^{-12}\exp(\frac{380}{T})$ | 2 |
| 24 | $EtOO + NO_3$ | $\rightarrow MeCHO + HO_2 + NO_2$ | $2.30\times10^{-12}$ | 2 |
| 25 | $EtOO + MeCO_3$ | $\rightarrow MeCHO + HO_2 + MeOO$ | $4.40\times10^{-13}\exp(\frac{1070}{T})$ | 2 |
| 26 | $MeCO_3 + NO$ | $\rightarrow MeOO + CO_2 + NO_2$ | $7.50\times10^{-12}\exp(\frac{290}{T})$ | 2 |
| 27 | $MeCO_3 + NO_3$ | $\rightarrow MeOO + CO_2 + NO_2$ | $4.00\times10^{-12}$ | 4 |
| 28 | $n\text{-}PrOO + NO$ | $\rightarrow EtCHO + HO_2 + NO_2$ | $2.90\times10^{-12}\exp(\frac{350}{T})$ | 2 |
| 29 | $n\text{-}PrOO + NO_3$ | $\rightarrow EtCHO + HO_2 + NO_2$ | $2.50\times10^{-12}$ | 4 |
| 30 | $i\text{-}PrOO + NO$ | $\rightarrow Me_2CO + HO_2 + NO_2$ | $2.70\times10^{-12}\exp(\frac{360}{T})$ | 2 |
| 31 | $i\text{-}PrOO + NO_3$ | $\rightarrow Me_2CO + HO_2 + NO_2$ | $2.50\times10^{-12}$ | 4 |
| 32 | $EtCO_3 + NO$ | $\rightarrow EtOO + CO_2 + NO_2$ | $6.70\times10^{-12}\exp(\frac{340}{T})$ | 2 |
| 33 | $EtCO_3 + NO_3$ | $\rightarrow EtOO + CO_2 + NO_2$ | $4.00\times10^{-12}$ | 4 |
| 34 | $MeCOCH_2OO + NO$ | $\rightarrow MeCO_3 + HCHO + NO_2$ | $2.80\times10^{-12}\exp(\frac{300}{T})$ | 5 |
| 35 | $MeCOCH_2OO + NO_3$ | $\rightarrow MeCO_3 + HCHO + NO_2$ | $2.50\times10^{-12}$ | 4 |
| 36 | $NO + NO_3$ | $\rightarrow NO_2 + NO_2$ | $1.80\times10^{-11}\exp(\frac{110}{T})$ | 2 |
| 37 | $NO + O_3$ | $\rightarrow NO_2$ | $1.40\times10^{-12}\exp(\frac{-1310}{T})$ | 2 |
| 38 | $NO_2 + O_3$ | $\rightarrow NO_3$ | $1.40\times10^{-13}\exp(\frac{-2470}{T})$ | 2 |





**Table 3 – continued from previous page**

| | Reactants | Products | $k$ | Reference |
|---|---|---|---|---|
| 39 | $NO_3 + HCHO$ | $\rightarrow HONO_2 + HO_2 + CO$ | $2.00\times10^{-12}\exp(\frac{-2440}{T})$ | 2 |
| 40 | $NO_3 + MeCHO$ | $\rightarrow HONO_2 + MeCO_3$ | $1.40\times10^{-12}\exp(\frac{-1860}{T})$ | 2 |
| 41 | $NO_3 + EtCHO$ | $\rightarrow HONO_2 + EtCO_3$ | $3.46\times10^{-12}\exp(\frac{-1862}{T})$ | 4 |
| 42 | $NO_3 + Me_2CO$ | $\rightarrow HONO_2 + MeCOCH_2OO$ | $3.00\times10^{-17}$ | 2 |
| 43 | $N_2O_5 + H_2O$ | $\rightarrow HONO_2 + HONO_2$ | $2.50\times10^{-22}$ | 2 |
| 44 | $O(^3P) + O_3$ | $\rightarrow O_2 + O_2$ | $8.00\times10^{-12}\exp(\frac{-2060}{T})$ | 2 |
| 45 | $O(^1D) + CH_4$ | $\rightarrow OH + MeOO$ | $1.05\times10^{-10}$ | 2 |
| 46 | $O(^1D) + CH_4$ | $\rightarrow HCHO + H_2$ | $7.50\times10^{-12}$ | 2 |
| 47 | $O(^1D) + CH_4$ | $\rightarrow HCHO + HO_2 + HO_2$ | $3.45\times10^{-11}$ | 2 |
| 48 | $O(^1D) + H_2O$ | $\rightarrow OH + OH$ | $2.20\times10^{-10}$ | 2 |
| 49 | $O(^1D) + N_2$ | $\rightarrow O(^3P) + N_2$ | $2.10\times10^{-11}\exp(\frac{115}{T})$ | 6 |
| 50 | $O(^1D) + O_2$ | $\rightarrow O(^3P) + O_2$ | $3.20\times10^{-11}\exp(\frac{67}{T})$ | 2 |
| 51 | $OH + CH_4$ | $\rightarrow H_2O + MeOO$ | $1.85\times10^{-12}\exp(\frac{-1690}{T})$ | 2 |
| 52 | $OH + C_2H_6$ | $\rightarrow H_2O + EtOO$ | $6.90\times10^{-12}\exp(\frac{-1000}{T})$ | 2 |
| 53 | $OH + C_3H_8$ | $\rightarrow n\text{-}PrOO + H_2O$ | $7.60\times10^{-12}\exp(\frac{-585}{T})$ | 2 |
| 54 | $OH + C_3H_8$ | $\rightarrow i\text{-}PrOO + H_2O$ | $7.60\times10^{-12}\exp(\frac{-585}{T})$ | 2 |
| 55 | $OH + CO$ | $\rightarrow HO_2$ | $1.44\times10^{-13}$ | 2 |
| 56 | $OH + EtCHO$ | $\rightarrow H_2O + EtCO_3$ | $5.10\times10^{-12}\exp(\frac{405}{T})$ | 2 |
| 57 | $OH + EtOOH$ | $\rightarrow H_2O + MeCHO + OH$ | $8.01\times10^{-12}$ | 4 |
| 58 | $OH + EtOOH$ | $\rightarrow H_2O + EtOO$ | $1.90\times10^{-12}\exp(\frac{190}{T})$ | 4 |
| 59 | $OH + H_2$ | $\rightarrow H_2O + HO_2$ | $7.70\times10^{-12}\exp(\frac{-2100}{T})$ | 2 |
| 60 | $OH + H_2O_2$ | $\rightarrow H_2O + HO_2$ | $2.90\times10^{-12}\exp(\frac{-160}{T})$ | 2 |
| 61 | $OH + HCHO$ | $\rightarrow H_2O + HO_2 + CO$ | $5.40\times10^{-12}\exp(\frac{135}{T})$ | 1 |
| 62 | $OH + HO_2$ | $\rightarrow H_2O$ | $4.80\times10^{-11}\exp(\frac{250}{T})$ | 2 |
| 63 | $OH + HO_2NO_2$ | $\rightarrow H_2O + NO_2$ | $1.90\times10^{-12}\exp(\frac{270}{T})$ | 2 |
| 64 | $OH + HO_2NO_2$ | $\rightarrow H_2O + NO_3$ | $1.50\times10^{-13}$ | 2 |
| 65 | $OH + HONO$ | $\rightarrow H_2O + NO_2$ | $2.50\times10^{-12}\exp(\frac{260}{T})$ | 2 |
| 66 | $OH + MeOOH$ | $\rightarrow H_2O + HCHO + OH$ | $1.02\times10^{-12}\exp(\frac{190}{T})$ | 2 |
| 67 | $OH + MeOOH$ | $\rightarrow H_2O + MeOO$ | $1.89\times10^{-12}\exp(\frac{190}{T})$ | 2 |
| 68 | $OH + MeONO_2$ | $\rightarrow HCHO + NO_2 + H_2O$ | $4.00\times10^{-13}\exp(\frac{-845}{T})$ | 2 |
| 69 | $OH + Me_2CO$ | $\rightarrow H_2O + MeCOCH_2OO$ | $8.80\times10^{-12}\exp(\frac{-1320}{T})$ | 2 |
| 70 | $OH + Me_2CO$ | $\rightarrow H_2O + MeCOCH_2OO$ | $1.70\times10^{-14}\exp(\frac{420}{T})$ | 2 |
| 71 | $OH + MeCOCH_2OOH$ | $\rightarrow H_2O + MeCOCH_2OO$ | $1.90\times10^{-12}\exp(\frac{190}{T})$ | 4 |
| 72 | $OH + MeCOCH_2OOH$ | $\rightarrow OH + MGLY$ | $8.39\times10^{-12}$ | 4 |
| 73 | $OH + MeCHO$ | $\rightarrow H_2O + MeCO_3$ | $4.40\times10^{-12}\exp(\frac{365}{T})$ | 2 |
| 74 | $OH + NO_3$ | $\rightarrow HO_2 + NO_2$ | $2.00\times10^{-11}$ | 2 |
| 75 | $OH + O_3$ | $\rightarrow HO_2 + O_2$ | $1.70\times10^{-12}\exp(\frac{-940}{T})$ | 2 |
| 76 | $OH + OH$ | $\rightarrow H_2O + O(^3P)$ | $6.31\times10^{-14}(\frac{T}{300})^{2.6}\exp(\frac{945}{T})$ | 2 |
| 77 | $OH + PAN$ | $\rightarrow HCHO + NO_2 + H_2O$ | $3.00\times10^{-14}$ | 2 |
| 78 | $OH + PPAN$ | $\rightarrow MeCHO + NO_2 + H_2O$ | $1.27\times10^{-12}$ | 4 |
| 79 | $OH + n\text{-}PrOOH$ | $\rightarrow n\text{-}PrOO + H_2O$ | $1.90\times10^{-12}\exp(\frac{190}{T})$ | 4 |





**Table 3 – continued from previous page**

| | Reactants | Products | $k$ | Reference |
|---|---|---|---|---|
| 80 | OH + n-PrOOH | → EtCHO + H$_2$O + OH | $1.10\times10^{-11}$ | 4 |
| 81 | OH + i-PrOOH | → i-PrOO + H$_2$O | $1.90\times10^{-12}\exp(\frac{190}{T})$ | 4 |
| 82 | OH + i-PrOOH | → Me$_2$CO + OH | $1.66\times10^{-11}$ | 4 |
| 83 | O($^3$P) + NO$_2$ | → NO + O$_2$ | $5.50\times10^{-12}\exp(\frac{188}{T})$ | 2 |
| 84 | OH + C$_5$H$_8$ | → ISO$_2$ | $2.70\times10^{-11}\exp(\frac{390}{T})$ | 2 |
| 85[a] | OH + C$_5$H$_8$ | → MACR + HCHO + MACRO$_2$ + MeCO$_3$ | $3.33\times10^{-15}\exp(\frac{-1995}{T})$ | 2 |
| 86[a] | OH + C$_5$H$_8$ | → MeOO + HCOOH + CO + H$_2$O$_2$ | $3.33\times10^{-15}\exp(\frac{-1995}{T})$ | 2 |
| 87[a] | OH + C$_5$H$_8$ | → HO$_2$ + OH | $3.33\times10^{-15}\exp(\frac{-1995}{T})$ | 2 |
| 88 | NO$_3$ + C$_5$H$_8$ | → ISON | $3.15\times10^{-12}\exp(\frac{-450}{T})$ | 2 |
| 89 | NO + ISO$_2$ | → NO$_2$ + MACR + HCHO + HO$_2$ | $2.43\times10^{-12}\exp(\frac{360}{T})$ | 4,7 |
| 90 | NO + ISO$_2$ | → ISON | $1.12\times10^{-13}\exp(\frac{360}{T})$ | 4,7 |
| 91 | HO$_2$ + ISO$_2$ | → ISOOH | $2.05\times10^{-13}\exp(\frac{1300}{T})$ | 4,7 |
| 92 | ISO$_2$ + ISO$_2$ | → MACR + MACR + HCHO + HO$_2$ | $2.00\times10^{-12}$ | 7 |
| 93 | OH + ISOOH | → MACR + OH | $1.00\times10^{-10}$ | 7 |
| 94 | OH + ISON | → HACET + NALD | $1.30\times10^{-11}$ | 7 |
| 95 | OH + MACR | → MACRO$_2$ | $1.30\times10^{-12}\exp(\frac{610}{T})$ | 2 |
| 96 | OH + MACR | → MACRO$_2$ | $4.00\times10^{-12}\exp(\frac{380}{T})$ | 2 |
| 97[a] | O$_3$ + MACR | → MGLY + HCOOH + HO$_2$ + CO | $2.13\times10^{-16}\exp(\frac{-1520}{T})$ | 2 |
| 98[a] | O$_3$ + MACR | → OH + MeCO$_3$ | $2.13\times10^{-16}\exp(\frac{-1520}{T})$ | 2 |
| 99[a] | O$_3$ + MACR | → MGLY + HCOOH + HO$_2$ + CO | $3.50\times10^{-16}\exp(\frac{-2100}{T})$ | 2 |
| 100[a] | O$_3$ + MACR | → OH + MeCO$_3$ | $3.50\times10^{-16}\exp(\frac{-2100}{T})$ | 2 |
| 101[a] | NO + MACRO$_2$ | → NO$_2$ + MeCO$_3$ + HACET + CO | $1.27\times10^{-12}\exp(\frac{360}{T})$ | 4,7 |
| 102[a] | NO + MACRO$_2$ | → MGLY + HCHO + HO$_2$ | $1.27\times10^{-12}\exp(\frac{360}{T})$ | 4,7 |
| 103 | HO$_2$ + MACRO$_2$ | → MACROOH | $1.83\times10^{-13}\exp(\frac{1300}{T})$ | 4,7 |
| 104[a] | MACRO$_2$ + MACRO$_2$ | → HACET + MGLY + HCHO + CO | $1.00\times10^{-12}$ | 4,7 |
| 105[a] | MACRO$_2$ + MACRO$_2$ | → HO$_2$ | $1.00\times10^{-12}$ | 4,7 |
| 106 | OH + MPAN | → HACET + NO$_2$ | $2.90\times10^{-11}$ | 2 |
| 107 | OH + MACROOH | → MACRO$_2$ | $3.00\times10^{-11}$ | 7 |
| 108 | OH + HACET | → MGLY + HO$_2$ | $3.00\times10^{-12}$ | 2,7 |
| 109 | OH + MGLY | → MeCO$_3$ + CO | $1.50\times10^{-11}$ | 2,7 |
| 110 | NO$_3$ + MGLY | → MeCO$_3$ + CO + HONO$_2$ | $3.46\times10^{-12}\exp(\frac{-1860}{T})$ | 4 |
| 111 | OH + NALD | → HCHO + CO + NO$_2$ | $4.40\times10^{-12}\exp(\frac{365}{T})$ | 2,7 |
| 112 | OH + MeCO$_3$H | → MeCO$_3$ | $3.70\times10^{-12}$ | 4,7 |
| 113 | OH + MeCO$_2$H | → MeOO | $4.00\times10^{-13}\exp(\frac{200}{T})$ | 8 |
| 114 | OH + HCOOH | → HO$_2$ | $4.50\times10^{-13}$ | 2 |
| 115 | MeOH + OH | → HCHO + HO$_2$ | $2.85\times10^{-12}\exp(\frac{-345}{T})$ | 3 |
| 116 | OH + C$_{10}$H$_{16}$ | → TERPO$_2$ | $1.20\times10^{-11}\exp(\frac{444}{T})$ | 9 |
| 117 | O$_3$ + C$_{10}$H$_{16}$ | → OH + MEK + HO$_2$ | $1.00\times10^{-15}\exp(\frac{-732}{T})$ | 9 |
| 118 | NO$_3$ + C$_{10}$H$_{16}$ | → ISON + MACR | $1.20\times10^{-12}\exp(\frac{490}{T})$ | 9 |
| 119[a] | NO + TERPO$_2$ | → Me$_2$CO + HO$_2$ + NO$_2$ | $2.10\times10^{-12}\exp(\frac{180}{T})$ | 9 |
| 120[a] | NO + TERPO$_2$ | → MACR + MACR | $2.10\times10^{-12}\exp(\frac{180}{T})$ | 9 |



**Table 3 – continued from previous page**

| | Reactants | Products | $k$ | Reference |
|---|---|---|---|---|
| 121[a] | $HO_2 + TERPO_2$ | $\rightarrow TERPOOH$ | $7.50\times10^{-13}\exp(\frac{700}{T})$ | 9 |
| 122[a] | $OH + TERPOOH$ | $\rightarrow TERPO_2$ | $3.80\times10^{-12}\exp(\frac{200}{T})$ | 9 |
| 123 | $C_4H_{10} + OH$ | $\rightarrow BtOO + H_2O$ | $9.10\times10^{-12}\exp(\frac{-405}{T})$ | 3 |
| 124[a] | $BtOO + NO$ | $\rightarrow NO_2 + MEK + HO_2 + EtOO$ | $1.27\times10^{-12}\exp(\frac{360}{T})$ | 4 |
| 125[a] | $BtOO + NO$ | $\rightarrow ONIT + MeCHO$ | $1.27\times10^{-12}\exp(\frac{360}{T})$ | 4 |
| 126 | $BtOO + HO_2$ | $\rightarrow BtOOH$ | $1.82\times10^{-13}\exp(\frac{1300}{T})$ | 4 |
| 127[a] | $BtOO + MeOO$ | $\rightarrow MEK + HCHO + HO_2 + MeCHO$ | $1.25\times10^{-13}$ | 4 |
| 128[a] | $BtOO + MeOO$ | $\rightarrow MeOH + EtOO$ | $1.25\times10^{-13}$ | 4 |
| 129[a] | $BtOOH + OH$ | $\rightarrow BtOO + MEK + OH + H_2O$ | $1.90\times10^{-12}\exp(\frac{190}{T})$ | 4 |
| 130 | $MEK + OH$ | $\rightarrow MEKOO$ | $1.30\times10^{-12}\exp(\frac{-25}{T})$ | 3 |
| 131 | $MEKOO + NO$ | $\rightarrow MeCHO + MeCO_3 + NO_2 + ONIT$ | $2.54\times10^{-12}\exp(\frac{360}{T})$ | 4 |
| 132 | $MEKOO + HO_2$ | $\rightarrow MEKOOH$ | $1.82\times10^{-13}\exp(\frac{1300}{T})$ | 4 |
| 133 | $MEKOOH + OH$ | $\rightarrow MeCOCOMe + OH + OH$ | $1.90\times10^{-12}\exp(\frac{190}{T})$ | 4 |
| 134 | $ONIT + OH$ | $\rightarrow MEK + NO_2 + H_2O$ | $1.60\times10^{-12}$ | 3 |
| 135[a] | $C_2H_4 + O_3$ | $\rightarrow HCHO + HO_2 + OH + CO$ | $4.55\times10^{-15}\exp(\frac{-2580}{T})$ | 3 |
| 136[a] | $C_2H_4 + O_3$ | $\rightarrow H_2 + CO_2 + HCOOH$ | $4.55\times10^{-15}\exp(\frac{-2580}{T})$ | 3 |
| 137[a] | $C_3H_6 + O_3$ | $\rightarrow HCHO + MeCHO + OH + HO_2$ | $1.83\times10^{-15}\exp(\frac{-1880}{T})$ | 3 |
| 138[a] | $C_3H_6 + O_3$ | $\rightarrow EtOO + MGLY + CH_4 + CO$ | $1.83\times10^{-15}\exp(\frac{-1880}{T})$ | 3 |
| 139[a] | $C_3H_6 + O_3$ | $\rightarrow MeOH + MeOO + HCOOH$ | $1.83\times10^{-15}\exp(\frac{-1880}{T})$ | 3 |
| 140[a] | $C_3H_6 + NO_3$ | $\rightarrow ONIT$ | $4.60\times10^{-13}\exp(\frac{-1155}{T})$ | 3 |
| 141[a] | $PrpeOO + NO$ | $\rightarrow MeCHO + HCHO + HO_2 + NO_2$ | $1.27\times10^{-12}\exp(\frac{360}{T})$ | 4 |
| 142[a] | $PrpeOO + NO$ | $\rightarrow ONIT$ | $1.27\times10^{-12}\exp(\frac{360}{T})$ | 4 |
| 143 | $PrpeOO + HO_2$ | $\rightarrow PrpeOOH$ | $1.50\times10^{-13}\exp(\frac{1300}{T})$ | 4 |
| 144 | $PrpeOOH + OH$ | $\rightarrow PrpeOO + H_2O$ | $1.90\times10^{-12}\exp(\frac{190}{T})$ | 4 |
| 145 | $PrpeOOH + OH$ | $\rightarrow HACET + OH$ | $2.44\times10^{-11}$ | 4 |
| 146[a] | $AROM + OH$ | $\rightarrow AROMO_2 + HO_2$ | $1.81\times10^{-12}\exp(\frac{338}{T})$ | 10 |
| 147[a] | $AROMO_2 + NO$ | $\rightarrow MGLY + NO_2 + MeCO_3 + CO$ | $1.35\times10^{-12}\exp(\frac{360}{T})$ | 10 |
| 148[a] | $AROMO_2 + NO$ | $\rightarrow HO_2$ | $1.35\times10^{-12}\exp(\frac{360}{T})$ | 10 |
| 149[a] | $AROMO_2 + NO_3$ | $\rightarrow MGLY + NO_2 + MeCO_3 + CO$ | $1.20\times10^{-12}$ | 10 |
| 150[a] | $AROMO_2 + NO_3$ | $\rightarrow HO_2$ | $1.20\times10^{-12}$ | 10 |
| 151[a] | $AROMO_2 + HO_2$ | $\rightarrow AROMOOH$ | $1.90\times10^{-13}\exp(\frac{-1300}{T})$ | 10 |
| 152[a] | $AROMO_2 + MeOO$ | $\rightarrow MGLY + CO + MeCO_3 + MeOH$ | $1.15\times10^{-13}$ | 10 |
| 153[a] | $AROMO_2 + MeOO$ | $\rightarrow HO_2 + HCHO$ | $1.15\times10^{-13}$ | 10 |
| 154[a] | $AROMOOH + OH$ | $\rightarrow AROMO_2$ | $1.90\times10^{-12}\exp(\frac{190}{T})$ | 10 |
| 155[a] | $AROMOOH + OH$ | $\rightarrow OH + H_2O$ | $4.61\times10^{-18}\exp(\frac{253}{T})$ | 10 |
| 156[a] | $AROMOOH + OH$ | $\rightarrow MeCO_3 + CO + HO_2 + OH$ | $4.19\times10^{-17}\exp(\frac{696}{T})$ | 10 |
| 157 | $HO_2 + O_3S$ | $\rightarrow HO_2 + O_2$ | $2.03\times10^{-16}(\frac{T}{300})^{4.57}\exp(\frac{693}{T})$ | 2 |
| 158 | $OH + O_3S$ | $\rightarrow OH + O_2$ | $1.70\times10^{-12}\exp(\frac{-940}{T})$ | 2 |
| 159 | $O(^1D)S + H_2O$ | $\rightarrow H_2O$ | $2.20\times10^{-10}$ | 2 |
| 160 | $O(^1D)S + N_2$ | $\rightarrow O(^3P)S + N_2$ | $2.10\times10^{-11}\exp(\frac{115}{T})$ | 6 |



**Table 3 – continued from previous page**

|     | Reactants | Products | $k$ | Reference |
|-----|-----------|----------|-----|-----------|
| 161 | $O(^1D)S + O_2$ | $\rightarrow O(^3P)S + O_2$ | $3.20 \times 10^{-11} \exp(\frac{67}{T})$ | 2 |

880

a: Reactions are split between multiple lines.





Table 4: TOMCAT gas-phase termolecular and thermal decomposition reactions. Rate constant $k = \left(\frac{k_0[M]}{1+k_0[M]/k_\infty}\right)F_c^{(1+[log\frac{k_0[M]}{k_\infty}]^2)^{-1}}$, where $k_0$ is the low pressure limit, $k_\infty$ is the high pressure limit and $M$ is the number density in molecules/cm$^3$. $F_c = f$ when $f < 1$ else $F_c = exp(-T/f)$. Low pressure limit $k_0 = k_1 \left(\frac{T}{300}\right)^{\alpha_1} exp\left(\frac{-\beta_1}{T}\right)$ and high pressure limit $k_\infty = k_2\left(\frac{T}{300}\right)^{\alpha_2} exp\left(\frac{-\beta_2}{T}\right)$. Reaction rate references 1: Atkinson et al. (b), 2: MCM (2004), 3: Pöschl et al. (2000), 4: Atkinson et al. (c).

| | Reactants | Products | $f$ | $k_1$ | $\alpha_1$ | $\beta_1$ | $k_2$ | $\alpha_2$ | $\beta_2$ | Reference |
|---|---|---|---|---|---|---|---|---|---|---|
| 1[a] | $HO_2 + HO_2 + M$ | $\rightarrow H_2O_2 + O_2 + M$ | 0.00 | $1.90\times10^{-33}$ | 0.00 | -980.0 | $0.00\times10^{+00}$ | 0.00 | 0.0[1] | 1 |
| 2 | $HO_2 + NO_2 + M$ | $\rightarrow HO_2NO_2 + M$ | 0.60 | $1.80\times10^{-31}$ | -3.20 | 0.0 | $4.70\times10^{-12}$ | 0.00 | 0.0 | 1 |
| 3 | $HO_2NO_2 + M$ | $\rightarrow HO_2 + NO_2 + M$ | 0.60 | $4.10\times10^{-05}$ | 0.00 | 10650.0 | $4.80\times10^{+15}$ | 0.00 | 11170.0 | 1 |
| 4 | $MeCO_3 + NO_2 + M$ | $\rightarrow PAN + M$ | 0.30 | $2.70\times10^{-28}$ | -7.10 | 0.0 | $1.20\times10^{-11}$ | -0.90 | 0.0 | 1 |
| 5 | $PAN + M$ | $\rightarrow MeCO_3 + NO_2 + M$ | 0.30 | $4.90\times10^{-03}$ | 0.00 | 12100.0 | $5.40\times10^{+16}$ | 0.00 | 13830.0 | 1 |
| 6 | $N_2O_5 + M$ | $\rightarrow NO_2 + NO_3 + M$ | 0.35 | $1.30\times10^{-03}$ | -3.50 | 11000.0 | $9.70\times10^{+14}$ | 0.10 | 11080.0 | 1 |
| 7 | $NO_2 + NO_3 + M$ | $\rightarrow N_2O_5 + M$ | 0.35 | $3.60\times10^{-30}$ | -4.10 | 0.0 | $1.90\times10^{-12}$ | 0.20 | 0.0 | 1 |
| 8 | $O(^3P) + O_2 + M$ | $\rightarrow O_3 + M$ | 0.00 | $5.70\times10^{-34}$ | -2.60 | 0.0 | $0.00\times10^{+00}$ | 0.00 | 0.0 | 1 |
| 9 | $OH + NO + M$ | $\rightarrow HONO + M$ | 1420.00 | $7.40\times10^{-31}$ | -2.40 | 0.0 | $3.30\times10^{-11}$ | -0.30 | 0.0 | 1 |
| 10 | $OH + NO_2 + M$ | $\rightarrow HONO_2 + M$ | 0.40 | $3.30\times10^{-30}$ | -3.00 | 0.0 | $4.10\times10^{-11}$ | 0.00 | 0.0 | 1 |
| 11 | $OH + OH + M$ | $\rightarrow H_2O_2 + M$ | 0.50 | $6.90\times10^{-31}$ | -0.80 | 0.0 | $2.60\times10^{-11}$ | 0.00 | 0.0 | 1 |
| 12 | $EtCO_3 + NO_2 + M$ | $\rightarrow PPAN + M$ | 0.30 | $2.70\times10^{-28}$ | -7.10 | 0.0 | $1.20\times10^{-11}$ | -0.90 | 0.0 | 2 |
| 13 | $PPAN + M$ | $\rightarrow EtCO_3 + NO_2 + M$ | 0.36 | $1.70\times10^{-03}$ | 0.00 | 11280.0 | $8.30\times10^{+16}$ | 0.00 | 13940.0 | 1 |
| 14 | $MACRO_2 + NO_2 + M$ | $\rightarrow MPAN + M$ | 0.30 | $2.70\times10^{-28}$ | 0.00 | 11280.0 | $8.30\times10^{+16}$ | 0.00 | 13940.0 | 3 |
| 15 | $MPAN + M$ | $\rightarrow MACRO_2 + NO_2 + M$ | 0.30 | $4.90\times10^{-03}$ | 0.00 | 12100.0 | $5.40\times10^{+16}$ | 0.00 | 13830.0 | 3 |
| 16 | $O(^3P) + O_2 + M$ | $\rightarrow O_3 + M$ | 0.00 | $5.70\times10^{-34}$ | -2.60 | 0.0 | $0.00\times10^{+00}$ | 0.00 | 0.0 | 1 |
| 17[b] | $C_2H_4 + OH + M$ | $\rightarrow PrpeOO + M$ | 0.48 | $2.87\times10^{-29}$ | -3.10 | 0.0 | $3.00\times10^{-12}$ | -0.85 | 0.0 | 4 |
| 18[b] | $C_2H_4 + OH + M$ | $\rightarrow PrpeOO + M$ | 0.48 | $2.87\times10^{-29}$ | -3.10 | 0.0 | $3.00\times10^{-12}$ | -0.85 | 0.0 | 4 |
| 19[b] | $C_2H_4 + OH + M$ | $\rightarrow$ | 0.48 | $2.87\times10^{-29}$ | -3.10 | 0.0 | $3.00\times10^{-12}$ | -0.85 | 0.0 | 4 |
| 20 | $C_3H_6 + OH + M$ | $\rightarrow PrpeOO + M$ | 0.50 | $8.00\times10^{-27}$ | -3.50 | 0.0 | $3.00\times10^{-11}$ | -1.00 | 0.0 | 4 |

a: Reaction rate is dependent on H$_2$O so k is weighted by factor of $1 + 1.4E - 21[H_2O]exp(2200/T)$, where [H$_2$O] is in molecules cm$^{-3}$.

b: Reactions are split between multiple lines.



Table 5: TOMCAT photolysis reactions.

| Reaction | Reactants | Products |
|---|---|---|
| 1 | EtOOH + h$\nu$ | $\rightarrow$ MeCHO + HO$_2$ + OH |
| 2 | H$_2$O$_2$ + h$\nu$ | $\rightarrow$ OH + OH |
| 3a | HCHO + h$\nu$ | $\rightarrow$ HO$_2$ + HO$_2$ + CO |
| 3b | HCHO + h$\nu$ | $\rightarrow$ H$_2$ + CO |
| 5 | HO$_2$NO$_2$ + h$\nu$ | $\rightarrow$ HO$_2$ + NO$_2$ |
| 6 | HONO$_2$ + h$\nu$ | $\rightarrow$ OH + NO$_2$ |
| 7a | MeCHO + h$\nu$ | $\rightarrow$ MeOO + HO$_2$ + CO |
| 7b | MeCHO + h$\nu$ | $\rightarrow$ CH4 + CO |
| 9 | MeOOH + h$\nu$ | $\rightarrow$ HO$_2$ + HCHO + OH |
| 10 | N$_2$O$_5$ + h$\nu$ | $\rightarrow$ NO$_3$ + NO$_2$ |
| 11 | NO$_2$ + h$\nu$ | $\rightarrow$ NO + O($^3$P) |
| 12a | NO$_3$ + h$\nu$ | $\rightarrow$ NO + O$_2$ |
| 12b | NO$_3$ + h$\nu$ | $\rightarrow$ NO$_2$ + O($^3$P) |
| 14 | O$_2$ + h$\nu$ | $\rightarrow$ O($^3$P) + O($^3$P) |
| 15a | O$_3$ + h$\nu$ | $\rightarrow$ O$_2$ + O($^1$D) |
| 15b | O$_3$ + h$\nu$ | $\rightarrow$ O$_2$ + O($^3$P) |
| 17 | PAN + h$\nu$ | $\rightarrow$ MeCO$_3$ + NO$_2$ |
| 18 | HONO + h$\nu$ | $\rightarrow$ OH + NO |
| 19 | EtCHO + h$\nu$ | $\rightarrow$ EtOO + HO$_2$ + CO |
| 20 | Me$_2$CO + h$\nu$ | $\rightarrow$ MeCO$_3$ + MeOO |
| 21 | n-PrOOH + h$\nu$ | $\rightarrow$ EtCHO + HO$_2$ + OH |
| 22 | i-PrOOH + h$\nu$ | $\rightarrow$ Me$_2$CO + HO$_2$ + OH |
| 23 | MeCOCH$_2$OOH + h$\nu$ | $\rightarrow$ MeCO$_3$ + HCHO + OH |
| 24 | PPAN + h$\nu$ | $\rightarrow$ EtCO$_3$ + NO$_2$ |
| 25 | MeONO$_2$ + h$\nu$ | $\rightarrow$ HO$_2$ + HCHO + NO$_2$ |
| 26a | TERPOOH + h$\nu$ | $\rightarrow$ OH + HO$_2$ + MACR + MACR |
| 26b | TERPOOH + h$\nu$ | $\rightarrow$ TERPOOH + Me$_2$CO |
| 28 | ISOOH + h$\nu$ | $\rightarrow$ OH + MACR + HCHO + HO$_2$ |
| 29 | ISON + h$\nu$ | $\rightarrow$ NO$_2$ + MACR + HCHO + HO$_2$ |
| 30 | MACR + h$\nu$ | $\rightarrow$ MeCO$_3$ + HCHO + CO + HO$_2$ |
| 31 | MPAN + h$\nu$ | $\rightarrow$ MACRO$_2$ + NO$_2$ |
| 32a | MACROOH + h$\nu$ | $\rightarrow$ OH + HO$_2$ + OH + HO$_2$ |
| 32b | MACROOH + h$\nu$ | $\rightarrow$ HACET + CO + MGLY + HCHO |
| 34 | HACET + h$\nu$ | $\rightarrow$ MeCO$_3$ + HCHO + HO$_2$ |
| 35 | MGLY + h$\nu$ | $\rightarrow$ MeCO$_3$ + CO + HO$_2$ + |
| 36 | NALD + h$\nu$ | $\rightarrow$ HCHO + CO + NO$_2$ + HO$_2$ |
| 37 | MeCO$_3$H + h$\nu$ | $\rightarrow$ MeOO + OH |
| 38a | O$_3$S + h$\nu$ | $\rightarrow$ O$_2$ + O(1D)S |
| 38b | O$_3$S + h$\nu$ | $\rightarrow$ O$_2$ + O(3P)S |
| 40a | BtOOH + h$\nu$ | $\rightarrow$ MEK + MEK + EtOO + MeCHO |



**Table 5 – continued from previous page**

| Reaction | Reactants | Products |
|---|---|---|
| 40b | BtOOH + h$\nu$ | $\rightarrow$ HO$_2$ + HO$_2$ |
| 40c | BtOOH + h$\nu$ | $\rightarrow$ OH + OH + OH |
| 43 | MEK + h$\nu$ | $\rightarrow$ MeCO$_3$ + EtOO |
| 44 | MeCOCOMe + h$\nu$ | $\rightarrow$ MeCO$_3$ + MeCO$_3$ |
| 45 | MEKOOH + h$\nu$ | $\rightarrow$ MeCO$_3$ + MeCHO + OH |
| 46a | ONIT + h$\nu$ | $\rightarrow$ NO$_2$ + MEK + HO$_2$ + EtOO |
| 46b | ONIT + h$\nu$ | $\rightarrow$ MeCHO + ONIT |
| 48a | AROMOOH + h$\nu$ | $\rightarrow$ OH + Me$_2$CO + HO$_2$ + CO |
| 48b | AROMOOH + h$\nu$ | $\rightarrow$ MeCO$_3$ + AROMOOH |

885