# Peer review of "The TOMCAT global chemical transport model v1.6: Description of chemical mechanism and model evaluation"

_Geoscientific Model Development, 2016_

## Short Comment (SC1) · 22 Aug 2016

Dear authors,

In my role as Executive editor of GMD, I would like to bring to your attention our Editorial version 1.1:

http://www.geosci-model-dev.net/8/3487/2015/gmd-8-3487-2015.html

This highlights some requirements of papers published in GMD, which is also available on the GMD website in the 'Manuscript Types' section:

http://www.geoscientific-model-development.net/submission/manuscript_types.html

In particular, please note that for your paper, the following requirement has not been met in the Discussions paper:

- "The main paper must give the model name and version number (or other unique identifier) in the title."

For a model evaluation it is important to know, which model version exactly was evaluated. Therefore, please add a version number for the TOMCAT model in the title upon your revised submission to GMD.

Yours,

Astrid Kerkweg

––––––––––––––––––––––––––––––––––

---

## Referee Comment (RC1) · Anonymous Referee #1 · 21 Sep 2016

Review of

"The TOMCAT global chemical transport model: Description of chemical mechanism and model evaluation" by Monks at al.

Overview:

The paper presents an evaluation of two 1-year runs with the latest version of the global TOMCAT CTM using observations and observation based climatologies for OH, O3, CO, NOy and some VOCs. The two runs are carried out with the same model version for two different years (2000 and 2008) using different emission data sets.

General remarks:

[Figure]
The presented evaluation of TOMCAT is reasonably comprehensive but the juxtaposition of the two runs is of little scientific value. Such an inter-comparison experiment, as any scientifically sound experiment, should only differ in the specific aspects, which are under investigation.

However, the two presented model runs differ in the year (2000 vs. 2008), in the biomass burning data set (GFED 3.1 vs. FINN), the anthropogenic emissions (ACCMIP vs Streets v1.2) and probably also in the prescribed CH4. Given the different years, not only the meteorology but also the biomass burning and VOC emissions will be different. It is not clear at all what reasonable conclusion can be drawn from the comparison of the two model runs.

As the paper reports on the update of the chemical mechanism, one would expect that two model runs with and without this update, which are otherwise identical (i.e. w.r.t emissions, year, meteorology, CH4) are compared. One should choose a year for which there are many observations available from campaigns or satellite observations.

The model description part is too short, in particular for the chemical mechanism. If this section – as the title suggest – is an important part of the paper, including only references or simply stating the fact of certain upgrades is not sufficient. It would be better to discuss the chemical mechanism in more detail and to give a motivation for the necessity and most importantly the impact of the upgrades.

Specific remarks:

P1 Abstract seems too long. It should be a single paragraph

P1 L18: The term "boundary conditions" is a bit confusing as you only change emissions. Better say "emissions" as deposition is not considered. The fixed CH4 can be described as "effective emissions"

P2 L25: As a CTM does not simulate winds, it seems better to say the transport is "driven" rather than constrained.

[Figure]

**GMDD**

P3 L4: Mention the motivations of the update. Introduce the choice of the setup of the different model runs you want to compare.

P3 L33: Aircraft emissions should be mentioned together with the rest of the emissions in section 2.2

P4 L12: Please provide more detail on the chemical scheme and motivate the several updates of the VOC chemistry. This would be of interest for the scientific community.

P4 L27: Please provide more detail, what uptake coefficients are used etc.

P5 L8: See my general comment. I think the two runs differ in too many and random aspect.

P5 L28: Explain how Streets differs from ACCMIP and what they have in common.

P5 L31: Are the biogenic emissions produced with exactly the same MEGAN model version and input.

P5 L33: Explain the differences between FINN and GFED.

P5 L34: It is not clear if the same method to constrain CH4 is used in both runs.

P6 L8: Please add an explanation how you tackle the problem that different years are evaluated. Mention more clearly when use timely data and when climatological data

P6 L39: Explain how you use the satellite for the different years. (GOME-2 was put in orbit only in 2006)

P7 L10: Should be "Surface data climatology"

P7 L30: Should be "Ozone data climatology"

P8 L 37: Did you verify the importance for the suggest pathway. It should result in higher OH values. The surface ozone concentrations also depend strongly on deposition. Is any dry deposition active over the ocean? Please comment how H2O is represented in the model.

P8 L 37: Please specify your choice of the tropopause

P9 L16-22: Consider putting the part on the observations in section 3

P10 L15: As you show the OH distribution of RUN_2000 and RUN_2008 please also discuss potential causes of the differences. Is the RUN_2000, which seems to have larger biases also worse in other aspects ? Are errors in OH related to errors in other species ?

P11 L14: Consider using the update of the MACCITY data suggest by Stein et al. (2014)

P11 L14: Consider also the role of transport.

P 11 L16: According to your Figure 4 there is too much OH in Tomcat in NH and SH in the lower troposphere.

P 11 L23: Do you discuss here RUN_2008 (?). In this run the FINN data and not GFAS is used. Please be more specific. Is the difference between FINN and GFED (table 1) due to the different years used or is there a systematic bias between FINN and GFAS?

P11 L33: Please specify what years do you use in the comparison. Note there is a CO trend of about 1%/yr (Worden et al. 2013)

P13 L34: It would be nice to also show the OH observations at Hohenpeissenberg. As far as I know OH is measured there as well.

P13 L42: Some comments on the assumed temporal profiles of the anthropogenic and biogenic VOC emissions would be interesting here.

P14 L8: The relation between emissions and concentrations seems a bit trivial if you do not specify in more detail, why the VOC emissions are different.

P14 L25: Please specify how r was calculated, i.e. using hourly, daily or monthly averages?

[Figure]

P14 L33: Is the overestimation over Europe also related to emissions ?

P 14 L38 Which biomass data set and year you are referring to? (see above)

P 15 (Summary)

P15 L1 -5: It would be great if the paper could show in any way the benefit of this upgrade. This is unfortunately not the case with the presented two model runs.

P16 L 19: Is there an explanation for the OH differences? Has the model upgrade improved the OH bias? Is the 2000 run generally worse because of the larger biases against Spivakovsky ?

Figures:

Figure 2 Use different colour range for O3 cross section to show more structures in the troposphere.

Figure 4. Rotate latitude vs CO graphs.

Figure 6 Which colour for which run?

Figure 9. Mark a-d) on panels. Consider including graph titles

---

## Referee Comment (RC2) · Anonymous Referee #2 · 26 Sep 2016

General comment

The manuscript concerns the description and validation of the TOMCAT Chemistry Transport Model. One point which is not clear is what significant model improvements has been made, compared to recent versions used for intercomparison studies, that warrants a new benchmarking study at this point in time. There has been the addition of higher Volatile Organic Compounds and new Isoprene/Monoterpene chemistry, with no significant updates to the dynamics, microphysics or transport in the CTM. Currently the manuscript is written as such that the reader has no idea as to whether these chemistry updates improve the model performance, where a 'before' and 'after' simulation for a chosen year is not provided. Two arbitrary simulations years are

presented (2000 and 2008), being run by different emissions and driven by different meteorology. Some of the validation is presented in a climatological sense, whereas some is presented for the simulation year in general, with some data being for different time periods. In general TOMCAT exhibits biases but the reasons given are not quantitative but rather speculative as the rigor of the analysis is not sufficient enough to provide concrete answers, with conclusions being used from previous studies presumably using an identical model version. One could exploit rarely used data such as from SAFARI, MINATROC and/or THESEO campaigns, which provides an unique snapshot across different chemical regimes if more data is necessary. In order to improve this manuscript I recommend a major revision following either (i) a focus on one year presumably the year 2000 using yearly specific measurements or (ii) do a sensitivity study with and without new chemistry for a chosen year to show effects and (hopefully) improvements.

Main comments

Are the 31 levels having a higher resolution in the troposphere? How many layers describe the UTLS and Stratosphere?

Is the meteorology smoothed between 6 hourly updates or are step edges applied to e.g. H2O fields?

Emission inventories: In order to attain the most accurate simulations using a CTM requires time evolving emission input data which e.g. captures the development in the transport sector. In TOMCAT aircraft NOx emissions come from the QUANTIFY project for the fixed year of 2002 which will surely underestimate the contributions in aircraft NOx over the simulation period (year 2008). A valid reason of using these fixed emissions is not given where e.g. the MACCity inventory provides yearly specific aircraft NOx estimates.

Are any burning heights applied for the biomass burning emissions?

Heterogeneous conversion: What is happening to the conversion of N2O5 into HNO3 on cloud surfaces? How do you calculate the available surface area density? Why is there only one heterogeneous reaction when HO2 conversion on aerosols is also now considered important?

Photolysis rates: How old is the absorption spectral data and quantum yields employed?. Many changes to the recommendations have been made since the 1980's but no details are given as to what improvements have been made. Photolysis drives tropospheric chemistry therefore an accurate description of diurnal variability is needed to capture e.g. good NO2 lifetimes. How are cloud and aerosol treated in terms of optical density through the column? Sun-spectrum used with any modifications due to the sun-cycles? As this is a benchmark paper it is not attractive that the reader has to trawl though the literature to get such details.

Given that the chemical scheme employed is for the description of tropospheric processes (e.g. no CFC's), how is the stratosphere constrained in order to get correct seasonality in the overhead O3 column and thus actinic flux profile?

Observations:

Why use GOME-2 and not GOME O3 data used for the year 2000 and 2008? Considering the current debate about the (potential) recovery of stratospheric O3 it appears the wrong decade is being used for validation. Why use an O3 sonde climatology from a completely different time period? This make little sense considering the choice of satellite data concerning O3. Both 2000 and 2008 annual means should be composed from the sondes and used to provide a better assessment of the vertical O3 profiles as the reader is unable to assess the ability of TOMCAT to capture inter-annual variability in the distributions. Were O3 measurements extracted on identical days for a valid comparison?

Validation:

Why lump NO and NO2 together?. Seasonality in NO2 mixing ratios is a proxy for the performance of the photolysis routine which is masked by lumping both oxides together. Even if there is a persistent bias the seasonality should be captured in some way.

Have the satellite comparisons for e.g. NO2 been done at the local overpass time for the days when measurements are available or in more ad-hoc way?

See comments above regarding O3 measurement data.

Specific comments:

Pg 8, ln 17: Providing seasonal means would be more informative for the reader, where significant changes should occur in the hemispheric zonal means. Pg 8, ln 45: Please introduce a table with the O3 diagnostics (Burden, Lifetime, Strat-Trop exchange and deposition) and then place in context using the new multi-model means. Also for the other gases such as CO and CH4, as information is scattered throughout the text. Pg 9, ln 40: Indeed, it would be even more illuminating by providing a Table of global mean OH from various studies. Pg 13, ln 32: Not surprising considering the mitigation practices over the last decade or two. This bias is potentially exaggerated by using non-yearly specific measurements. Many studies have focused on VOC emissions so can the authors state whether this is an artifact of CTM's in general? Pg 15, ln 15: Looking at Figure 17 I can see discrepencies of >100 percent for some of the points. Better to discuss seasonality to identify which season has the largest bias given that PAN is temperature sensitive. Pg 15, ln 15: " . . . may be too high . . .". If a comparison has been made then surely it either is or isn't too high for this period. Pg 15, ln 19: What is the relevance of GEOS-chem to this paper? What about the multitude of other CTM's? Figure 14 and 15: What is the motivation for lumping the measurements but not the model results? There are trends in surface observations related to emission trends. Is the bias larger for 2008 than 2000 i.e. does the deviance increase with time due to incorrect emission estimates?

---

## Author Comment (AC1) · 5 Dec 2016

Author Response to Reviewer Comments (RC1)

The authors would like to thank the reviewer for taking the time to read and comment on the submitted manuscript. The comments are repeated below (in quotation marks) followed by our responses. We have taken the comments on board and addressed them as described.

"The presented evaluation of TOMCAT is reasonably comprehensive but the juxtaposition of the two runs is of little scientific value. Such an inter-comparison experiment, as any scientifically sound experiment, should only differ in the specific aspects, which are

under investigation. However, the two presented model runs differ in the year (2000 vs. 2008), in the biomass burning data set (GFED 3.1 vs. FINN), the anthropogenic emissions (ACCMIP vs Streets v1.2) and probably also in the prescribed CH4. Given the different years, not only the meteorology but also the biomass burning and VOC emissions will be different. It is not clear at all what reasonable conclusion can be drawn from the comparison of the two model runs. As the paper reports on the update of the chemical mechanism, one would expect that two model runs with and without this update, which are otherwise identical (i.e. w.r.t emissions, year, meteorology, CH4) are compared. One should choose a year for which there are many observations available from campaigns or satellite observations. The model description part is too short, in particular for the chemical mechanism. If this section – as the title suggest – is an important part of the paper, including only references or simply stating the fact of certain upgrades is not sufficient. It would be better to discuss the chemical mechanism in more detail and to give a motivation for the necessity and most importantly the impact of the upgrades."

The main purpose of this paper is to document the version of the TOMCAT chemical mechanism that is now being used for scientific studies. The current chemical mechanism has only been previously documented in a PhD thesis (Monks, 2011). This means that users of the TOMCAT model would benefit from the chemical mechanism being documented in a peer-reviewed journal which is easily accessible. The earlier version of the chemistry scheme described by Arnold et al., (2005) is now obsolete and we see little value in showing an evaluation of this old scheme. However, we have added in a short discussion to Section 2.1 (Section 2.1.1 Impacts of extended VOC chemistry), which describes the impact of changing the chemistry from Arnold et al., (2005) to the current scheme. These results have been taken from Monks (2011) and are used to show the change in CO, O3 and OH. The description of the chemical mechanism has also been extended. The overall changes are an increase in CO and O3 and a decrease in OH.

On consideration of the reviewer comments, we agree that substantial confusion arises from the inclusion of both the RUN_2000 and RUN_2008 simulations, and so RUN_2000 has been removed from the paper. Our original intention was to show the two separate RUN_2000 and RUN_2008 simulations in order to provide best representation of the year 2000 and 2008, respectively, and make use of different datasets that are more representative of early years (e.g. the aircraft climatology) and later years (satellite and surface data). We believed that this would be useful in identifying any errors that may exist in the model that are not systematic and may be confined to the set-up for the particular year. This was useful in showing species like PAN and NOx are very sensitive to how the model is set-up. However, since these simulations use very different emissions data and meteorology that differs according to year, as highlighted, it is difficult to know the exact cause of differences in the model (meteorology versus emissions). We have also removed comparisons to the aircraft climatology (data for 1992-2001) as this dataset was more comparable to RUN_2000. We have kept RUN_2008 and have performed some additional comparisons to the 2008 POLARCAT campaign aircraft data to provide some data for other VOCs and NOy species that are not regularly measured. The surface data is now also shown for the year 2008 only.

Specific remarks:

"P1 Abstract seems too long. It should be a single paragraph." The abstract has been shortened.

"P1 L18: The term "boundary conditions" is a bit confusing as you only change emissions. Better say "emissions" as deposition is not considered. The fixed CH4 can be described as "effective emissions"" Sentence no longer applies as RUN_2000 has been removed.

"P2 L25: As a CTM does not simulate winds, it seems better to say the transport is "driven" rather than constrained." Changed to driven.

"P3 L4: Mention the motivations of the update. Introduce the choice of the setup of

the different model runs you want to compare." Changed to: "This paper summarises the latest tropospheric chemical mechanism scheme used in TOMCAT (Section 2). The scheme gives a more detailed representation of hydrocarbon chemistry than previously included in the model, with the inclusion of the oxidation of ethene, propene, butane, toluene and monoterpenes. Alkenes have the greatest potential for forming $O_3$ (Saunders et al., 2003), and previously, isoprene was the only alkene treated in the TOMCAT model. In addition, a more extensive VOC scheme makes it possible to couple the TOMCAT tropospheric chemistry to the formation of secondary organic aerosol in future versions of the GLOMAP aerosol model (Mann et al., 2010)."

"P3 L33: Aircraft emissions should be mentioned together with the rest of the emissions in section 2.2" This has been moved.

"P4 L12: Please provide more detail on the chemical scheme and motivate the several updates of the VOC chemistry. This would be of interest for the scientific community." This has been extended: "
[revised manuscript text omitted]

Archibald, A. T., Jenkin, M. E., and Shallcross, D. E.: An isoprene mechanism intercomparison, Atmos. Environ., 44, 5356–5364, doi:10.1016/j.atmosenv.2009.09.016, 2010. Archibald, A. T., Levine, J. G., Abraham, N. L., Cooke, M. C., Edwards, P. M., Heard, D. E., Jenkin, M. E., Karunaharan, A., Pike, R. C., Monks, P. S., Shallcross, D. E., Telford, P. J., Whalley, L. K., and Pyle, J. A.: Impacts of HOx regeneration and recycling in the oxidation of isoprene: consequences for the composition of past, present and future atmospheres, Geophys. Res. Lett., 38, L05804, doi:10.1029/2010GL046520, 2011. Atkinson, R. and Arey, J.: Gas-phase tropospheric chemistry of biogenic volatile organic compounds: a review, Atmospheric Environment, Volume 37, Supplement 2, 2003, Pages 197-219, ISSN 1352-2310, http://dx.doi.org/10.1016/S1352-2310(03)00391-1. Atkinson, R., Baulch, D. L., Cox, R. A., Crowley, J. N., Hampson, R. F., Hynes, R. G., Jenkin, M. E., Rossi, M. J., and Troe, J.: Evaluated kinetic and photochemical data for atmospheric chemistry: Volume I - gas phase reactions of Ox, HOx, NOxÂăand SOxÂăspecies, Atmos. Chem. Phys., 4, 1461-1738, doi:10.5194/acp-4-1461-2004, 2004. Atkinson, R., Baulch, D. L., Cox, R. A., Crowley, J. N., Hampson, R.

F., Hynes, R. G., Jenkin, M. E., Rossi, M. J., Troe, J., and IUPAC Subcommittee: Evaluated kinetic and photochemical data for atmospheric chemistry: Volume II – gas phase reactions of organic species, Atmos. Chem. Phys., 6, 3625-4055, doi:10.5194/acp-6-3625-2006, 2006. Chameides, W., Lindsay, R., Richardson, J., and Kiang, C.: The role of biogenic hydrocarbons in urban photochemical smog – Atlanta as a case-study, Science, 241, 1473–1475, 1988. Chipperfield, M.P., S.S. Dhomse, W. Feng, R.L. McKenzie, G. Velders and J.A. Pyle, Quantifying the ozone and UV benefits already achieved by the Montreal Protocol, Nature Communications, 6, 7233, doi:10.1038/ncomms8233, 2015. Fuentes, J.D., M. Lerdau, R. Atkinson, D. Baldocchi, J.W. Bottenheim, P. Ciccioli, B. Lamb, C. Geron, L. Gu, A. Guenther, T.D. Sharkey, W. Stockwell: Biogenic hydrocarbons in the atmospheric boundary layer: a review, Bulletin of the American Meteorological Society, 81 (2000), pp. 1537–1575 Guenther, A., Karl, T., Harley, P., Wiedinmyer, C., Palmer, P. I., and Geron, C.: Estimates of global terrestrial isoprene emissions using MEGAN (Model of Emissions of Gases and Aerosols from Nature), Atmos. Chem. Phys., 6, 3181-3210, doi:10.5194/acp-6-3181-2006, 2006. von Kuhlmann, R.: Photochemistry of Tropospheric Ozone, its Precursors and the Hydroxyl Radical: A 3D-Modeling Study Considering Non-Methane Hydrocarbons, Ph.D. thesis, Johannes Gutenberg-Universitat Mainz, Mainz, Germany, 2001. Saunders, S. M., Jenkin, M. E., Derwent, R. G., and Pilling, M. J.: Protocol for the development of the Master Chemical Mechanism, MCM v3 (Part A): tropospheric degradation of nonaromatic volatile organic compounds, Atmos. Chem. Phys., 3, 161–180, doi:10.5194/acp-3-161-2003, 2003. Squire, O. J., Archibald, A. T., Griffiths, P. T., Jenkin, M. E., Smith, D., and Pyle, J. A.: Influence of isoprene chemical mechanism on modelled changes in tropospheric ozone due to climate and land use over the 21st century, Atmos. Chem. Phys., 15, 5123-5143, doi:10.5194/acp-15-5123-2015, 2015. Sander, S.P., et al.: Chemical Kinetics and Photochemical Data for Use in Atmospheric Studies Evaluation Number 17. JPL Publication 10-6, Jet Propulsion Laboratory, Pasadena, USA, 2011. Sander, S.P., et al.: Chemical Kinetics and Photochemical Data for Use in Atmospheric Studies Evaluation Number 15. JPL Publication 06-2, Jet Propulsion Laboratory, Pasadena,

USA, 2006. Sukhodolov, T., E. Rozanov, W.T. Ball, A. Bais, K. Tourpali, A.I. Shapiro, P. Telford, S. Smyshlyaev, B. Fomin, R. Sander, S. Bossay, S. Bekki, M. Marchand, M.P. Chipperfield, S. Dhomse, J.D. Haigh, T. Peter and W. Schmutz, Evaluation of the simulated photolysis rates and their response to solar irradiance variability, J. Geophys. Res., 121, 6066-6084, doi:10.1002/2015JD024277, 2016. Wang, K. and Shallcross, D.: Modelling terrestrial biogenic isoprene fluxes and their potential impact on global chemical species using a coupled LSM-CTM model, Atmos. Environ., 34, 2909–2925, 2000.

"P4 L27: Please provide more detail, what uptake coefficients are used etc." We have added a table with the uptake coefficients.

"P5 L8: See my general comment. I think the two runs differ in too many and random aspect." RUN_2000 has now been removed to avoid confusion.

"P5 L28: Explain how Streets differs from ACCMIP and what they have in common." The description of Streets has been extended.

"P5 L31: Are the biogenic emissions produced with exactly the same MEGAN model version and input." Changed to: "Monthly mean biogenic emissions are from the MACC (Monitoring Atmospheric Composition and Climate) project (MACCity), which provides simulated VOCs calculated offline by the Model of Emissions of Gases and Aerosols from Nature (MEGAN) v2.0 for a reference year 2000 (Guenther et al., 2006)."

"P5 L33: Explain the differences between FINN and GFED." No longer relevant.

"P5 L34: It is not clear if the same method to constrain CH4 is used in both runs." This no longer relevant as RUN_2000 has been removed. A description of the two different ways of constraining methane is given earlier.

"P6 L8: Please add an explanation how you tackle the problem that different years are evaluated. Mention more clearly when use timely data and when climatological data" No longer relevant. RUN_2000 has been removed and RUN_2008 is compared to year

2008 observations. Yearly specific observations are now used.

"P6 L39: Explain how you use the satellite for the different years. (GOME-2 was put in orbit only in 2006)" Satellite O3 data was only compared to RUN_2008 as GOME-2 was only launched in 2006. RUN_2000 has been removed so this should no longer be confusing.

"P7 L10: Should be "Surface data climatology"" No longer relevant. We use only 2008 observations.

"P7 L30: Should be "Ozone data climatology"" Changed.

"P8 L 37: Did you verify the importance for the suggest pathway. It should result in higher OH values. The surface ozone concentrations also depend strongly on deposition. Is any dry deposition active over the ocean? Please comment how H2O is represented in the model." No. This would be interesting to look at in the future. We have removed this explanation. Yes, O3 deposition does occur over the ocean and water vapour is now described in the model description section.

"P8 L 37: Please specify your choice of the tropopause." We have not used a tropopause here as we are simply discussing the general distribution of high/low ozone. We have changed the sentence to be clearer: "In the tropics, lower O3 concentrations are seen at 100-300 hPa due to a higher tropopause in this region and the uplift of air with low O3 within deep tropical convection. At around 20-40 S/N, evidence of the downward transport of stratospheric O3 by the Brewer-Dobson circulation (Butchart, 2014) can be seen.

"P9 L16-22: Consider putting the part on the observations in section 3." Moved.

"P10 L15: As you show the OH distribution of RUN_2000 and RUN_2008 please also discuss potential causes of the differences. Is the RUN_2000, which seems to have larger biases also worse in other aspects? Are errors in OH related to errors in other species?" RUN_2000 is no longer included.

"P11 L14: Consider using the update of the MACCITY data suggest by Stein et al. (2014)" Whilst Stein et al., (2014) do show that increases in MACCity transportation emissions improve comparisons with some observations, they also show a similar performance when using the RETRO/REAS emissions. Streets emissions also uses REAS over Asia and other best estimates of emissions when regional inventories exist, with EDGAR to fill in the gaps. These emissions are a best guess for the year 2008 and were produced for the POLARCAT campaign. As we are using the POLARCAT data for comparison to RUN_2008, we have chosen to retain the Streets anthropogenic emissions.

"P11 L14: Consider also the role of transport." Added sentence saying: "Transport errors in the model could also play a role, however, they are unlikely to cause such widespread biases of this magnitude."

"P 11 L16: According to your Figure 4 there is too much OH in Tomcat in NH and SH in the lower troposphere." We have changed the sentence to reflect this: "In addition to this, as mentioned in Section 4.2, OH in the TOMCAT model is most likely too high at the surface, particularly in the tropics, and the NH/SH OH ratio is higher than estimates based on observations. This is likely to influence the lifetime of simulated CO and will contribute to the NH and SH biases."

"P 11 L23: Do you discuss here RUN_2008 (?). In this run the FINN data and not GFAS is used. Please be more specific. Is the difference between FINN and GFED (table 1) due to the different years used or is there a systematic bias between FINN and GFAS?" Yes, this is RUN_2008 with FINN data. RUN_2000 has been removed so there should be no more confusion in terms of which emissions are being used.

"P11 L33: Please specify what years do you use in the comparison. Note there is a CO trend of about 1%/yr (Worden et al. 2013)." MOPITT 2008 data is used to compare to RUN_2008. It was previously noted in the Figure 4 caption, however, this has been made more clear in the text.

"P13 L34: It would be nice to also show the OH observations at Hohenpeissenberg. As far as I know OH is measured there as well." We have been unable to locate any OH measurements for the year 2008. There were a few campaigns where measurements were made intensively and there was also some data that existed over a five-year period 1999-2003. We chose not to pursue this any further as we are not sure what this would add to the paper. It would be difficult to use them to evaluate the model due to OH being very short-lived and the model grid box being so large.

"P13 L42: Some comments on the assumed temporal profiles of the anthropogenic and biogenic VOC emissions would be interesting here." This is described in Section 2.2 so the reader could find out the emission temporal resolution here. As the measurements are made continuously and we are using monthly means to compare to the model we are not sure why this should be added to the discussion. If we were using observation made at specific time of the day, then yes this would be worth discussing.

"P14 L8: The relation between emissions and concentrations seems a bit trivial if you do not specify in more detail, why the VOC emissions are different." We are unable to find P14 L8. We assume that the comment refers to a difference in concentrations between RUN_2000 and RUN_2008. RUN_2000 is no longer included so we believe this comment no longer applies.

"P14 L25: Please specify how r was calculated, i.e. using hourly, daily or monthly averages?" The correlations are calculated between the observations and the model data shown in Figure 15. This is monthly mean data. We have added this information into the figure caption.

"P14 L33: Is the overestimation over Europe also related to emissions?" Yes, it is most likely linked to the emissions. The largest biases occur over the Baltic and North Seas where ship emissions are located. This could indicate a positive bias in these emissions.

"P 14 L38 Which biomass data set and year you are referring to? (see above)." We

are referring to FINN fire emissions as this is the RUN_2008 comparison. We have clarified this. Also, now RUN_2000 has been removed there will be less confusion.

"P15 L1 -5: It would be great if the paper could show in any way the benefit of this upgrade. This is unfortunately not the case with the presented two model runs." This is now discussed with the addition of a section showing the difference between two simulations with and without the extended chemistry. The new chemistry slightly increases CO in the model and therefore reduces the model negative bias. Global mean OH is reduced slightly to lower the positive bias compared with estimates based on methyl chloroform. Ozone is also higher due to the extra VOC emissions in the extended chemistry version of the model.

"P16 L 19: Is there an explanation for the OH differences? Has the model upgrade improved the OH bias? Is the 2000 run generally worse because of the larger biases against Spivakovsky?" No, it is still not fully understood why models and observations show this difference in the NH/SH OH ratio. We have expanded on this in the text. The model global mean OH is slightly lower with the extended chemistry, so yes this does improve the bias. RUN_2000 has been removed.

"Figure 2 Use different colour range for O3 cross section to show more structures in the troposphere." Done.

"Figure 4. Rotate latitude vs CO graphs." While we understand it seems to make more sense that the y-axis is CO in terms of it being the dependent variable, we have chosen to keep the figure as it is so you can compare c to a/b and f to d/e by eye.

"Figure 6 Which colour for which run?" RUN_2000 has been removed. Also, the label is already shown.

"Figure 9. Mark a-d) on panels. Consider including graph titles." Done.

---

## Author Comment (AC2) · 5 Dec 2016

Author Response to Reviewer Comments (RC2)

The authors would like to thank the reviewer for taking the time to read and comment on the submitted manuscript. The comments are repeated below (in quotation marks) followed by our responses. We have taken the comments on board and addressed them as described.

"The manuscript concerns the description and validation of the TOMCAT Chemistry Transport Model. One point which is not clear is what significant model improvements has been made, compared to recent versions used for intercomparison studies, that

warrants a new benchmarking study at this point in time. There has been the addition of higher Volatile Organic Compounds and new Isoprene/Monoterpene chemistry, with no significant updates to the dynamics, microphysics or transport in the CTM. Currently the manuscript is written as such that the reader has no idea as to whether these chemistry updates improve the model performance, where a 'before' and 'after' simulation for a chosen year is not provided. Two arbitrary simulations years are presented (2000 and 2008), being run by different emissions and driven by different meteorology. Some of the validation is presented in a climatological sense, whereas some is presented for the simulation year in general, with some data being for different time periods. In general TOMCAT exhibits biases but the reasons given are not quantitative but rather speculative as the rigor of the analysis is not sufficient enough to provide concrete answers, with conclusions being used from previous studies presumably using an identical model version. One could exploit rarely used data such as from SAFARI, MINATROC and/or THESEO campaigns, which provides an unique snapshot across different chemical regimes if more data is necessary. In order to improve this manuscript I recommend a major revision following either (i) a focus on one year presumably the year 2000 using yearly specific measurements or (ii) do a sensitivity study with and without new chemistry for a chosen year to show effects and (hopefully) improvements."

The main purpose of this paper is to document the version of the TOMCAT chemical mechanism that is now being used for scientific studies. Whilst this scheme has been used in some recent scientific studies (Emmons et al., 2015 and Richards et al., (2013), the current chemical mechanism has only been previously documented in a PhD thesis (Monks, 2011). Emmons et al., (2016) and Richards et al., (2013) only compared the model to data in limited regions and over limited time periods. Therefore, a benchmarking paper is warranted, where evaluation of key components can be made globally throughout the year. Users of the TOMCAT model and GLOMAP-TOMCAT coupled model would also benefit from the chemical mechanism being documented as the only other published description of the TOMCAT chemistry scheme is described by Arnold

et al., (2005), which is now obsolete. We see little value in showing an evaluation of this old scheme due to it not being used anymore, and therefore only evaluate the current scheme. However, a short discussion has been added to Section 2.1 (Section 2.1.1 Impacts of additional VOC chemistry) to describe the impact of changing the chemistry from Arnold et al., (2005) to the current scheme. These results have been taken from Monks (2011) and show the change in CO, O3 and OH. The overall changes are an increase in the burdens of CO and O3 and a decrease in OH. As with all models, TOM-CAT is negatively biased in CO, so an increase in the CO burden reduces this bias for the TOMCAT model. Simulated global mean OH is also higher than estimates calculated from methyl chloroform, therefore a decrease in OH provides a better simulation of OH.

Following similar comments from Reviewer 1 on confusion caused by presenting the two simulations from two different years, we have chosen to remove the year 2000 simulation. The aircraft climatology has also been removed and we show surface and aircraft comparisons for the year 2008 only. Please see our response to Reviewer 1 for more details.

Main comments: "Are the 31 levels having a higher resolution in the troposphere? How many layers describe the UTLS and Stratosphere?" The levels have a higher resolution in the boundary layer/lower troposphere and near the UTLS. A figure has been added to the manuscript to show this.

"Is the meteorology smoothed between 6 hourly updates or are step edges applied to e.g. H2O fields?" The meteorological data is linearly interpolated between the 6-hour updates. This information has been added to the manuscript.

"Emission inventories: In order to attain the most accurate simulations using a CTM requires time evolving emission input data which e.g. captures the development in the transport sector. In TOMCAT aircraft NOx emissions come from the QUANTIFY project for the fixed year of 2002 which will surely underestimate the contributions in

aircraft NOx over the simulation period (year 2008). A valid reason of using these fixed emissions is not given where e.g. the MACCity inventory provides yearly specific aircraft NOx estimates." I was not able to find any yearly specific MACCity aircraft emissions to evaluate this discrepancy properly. As MACCity emissions are simply a linear interpolation between the data I am using and the RCP 8.5 2005 and 2010 estimates, I looked at the RCP 8.5 emission dataset to assess by how much emissions have changed. Annual aircraft emissions of NOx are estimated to have increased from 2.8 Tg/yr to 3.1 Tg/yr between 2000 and 2010 (10 –year period). Assuming a linear increase, this would be ∼0.2 Tg (7%) change in aircraft emissions in the model over a 7-year period. We feel that the impact of using 2002 estimates instead of 2008 estimates will only cause a maximum difference of a few ppbv in O3 in the UT. This is because simulations with and without aircraft emissions have shown a maximum of 7 ppbv difference during the summer (Gauss et al., 2006). For this reason, we feel that this does not warrant a change in the emissions at this time but will be worth changing them in the future when the emissions are being updated.

"Are any burning heights applied for the biomass burning emissions?" No.

"Heterogeneous conversion: What is happening to the conversion of N2O5 into HNO3 on cloud surfaces? How do you calculate the available surface area density? Why is there only one heterogeneous reaction when HO2 conversion on aerosols is also now considered important?" There is no treatment of N2O5 uptake on cloud surfaces currently due to the use of climatological clouds in the model. This is something that will be considered in future versions of the model. Surface area density is calculated from offline simulated aerosol radius and number density for the 5 different aerosol types that are described in the paper. HO2 uptake onto aerosol is indeed important. The code exists for this to be included in the future but it has not been tested currently and is therefore not implemented in this version of the model. These points have been clarified in the manuscript.

"Photolysis rates: How old is the absorption spectral data and quantum yields employed?. Many changes to the recommendations have been made since the 1980's but no details are given as to what improvements have been made. Photolysis drives tropospheric chemistry therefore an accurate description of diurnal variability is needed to capture e.g. good NO2 lifetimes. How are cloud and aerosol treated in terms of optical density through the column? Sun-spectrum used with any modifications due to the sun-cycles? As this is a benchmark paper it is not attractive that the reader has to trawl though the literature to get such details." The photolysis scheme for the tropospheric chemistry scheme described here shares the 'library' of photochemical data with the stratospheric scheme (not discussed) and is regularly updated. Sun-cycles are not accounted for. More details of the model photolysis scheme have been added into the revised paper (see also response to Reviewer 1).

"Given that the chemical scheme employed is for the description of tropospheric processes (e.g. no CFC's), how is the stratosphere constrained in order to get correct seasonality in the overhead O3 column and thus actinic flux profile?" There is no stratospheric chemistry in this version of the model. We constrain stratospheric O3 with offline generated fields from the Cambridge 2-D model. A description of this has been added in Section 2.1 (also see RC1).

"Why use GOME-2 and not GOME O3 data used for the year 2000 and 2008? Considering the current debate about the (potential) recovery of stratospheric O3 it appears the wrong decade is being used for validation. Why use an O3 sonde climatology from a completely different time period? This make little sense considering the choice of satellite data concerning O3. Both 2000 and 2008 annual means should be composed from the sondes and used to provide a better assessment of the vertical O3 profiles as the reader is unable to assess the ability of TOMCAT to capture inter-annual variability in the distributions. Were O3 measurements extracted on identical days for a valid comparison?" We have removed RUN_2000 so using GOME data is no longer required. The GOME Ozone record experienced a problem where the tape recorder failed on the ERS-2 satellite onÂă22nd June 2003. Therefore, data could only be

download while it was within direct line of sight of a ground station resulting in reduced spatial. GOME-2 has global coverage for 2008 with improvements in the resolution and quality of the data. That is why we chose to retain RUN_2008.

"Why lump NO and NO2 together?. Seasonality in NO2 mixing ratios is a proxy for the performance of the photolysis routine which is masked by lumping both oxides together. Even if there is a persistent bias the seasonality should be captured in some way." This has been changed so we now show NO2 comparisons.

"Have the satellite comparisons for e.g. NO2 been done at the local overpass time for the days when measurements are available or in more ad-hoc way?" TOMCAT composition data have been co-located in time and space to both the GOME-2 Ozone and OMI NO2 products. Here the closest model grid box to the satellite pixel is sampled, within 3 hours of the satellite daytime overpass (e.g. 13.30 LT for OMI) as the model output is every 6 hours.Âă

Specific comments: "Pg 8, ln 17: Providing seasonal means would be more informative for the reader, where significant changes should occur in the hemispheric zonal means. " We have chosen to show annual means so they can be compared to the papers that are referenced in the discussion (Young et al., 2013; Voulgarakis et al., 2013), which show annual mean concentrations of O3 and OH from the mean of multiple models. Seasonal O3 and CO are shown in comparisons to observational data.

"Pg 8, ln 45: Please introduce a table with the O3 diagnostics (Burden, Lifetime, Strat-Trop exchange and deposition) and then place in context using the new multi-model means. Also for the other gases such as CO and CH4, as information is scattered throughout the text." A table has been added for the O3 burden, global mean OH concentration and CH4 lifetime. Some O3 diagnostics (strat-trop exchange and deposition) have not been calculated as diagnostics from these experiments unfortunately.

"Pg 9, ln 40: Indeed, it would be even more illuminating by providing a Table of global mean OH from various studies." A table has been added.

"Pg 13, ln 32: Not surprising considering the mitigation practices over the last decade or two. This bias is potentially exaggerated by using non-yearly specific measurements. Many studies have focused on VOC emissions so can the authors state whether this is an artifact of CTM's in general?" These comparisons are no longer included. ARCTAS 2008 data and 2008 surface data show a similar discrepancy, therefore, it is not due to a mismatch between emissions in the model and the years that the observations were made. This appears to be an artifact in CTMs in general. The paper already referenced (Emmons et al., 2015) in the discussion shows that this is widespread across different models with different OH concentrations so it likely due to underestimated emissions.

"Pg 15, ln 15: Looking at Figure 17 I can see discrepencies of >100 percent for some of the points. Better to discuss seasonality to identify which season has the largest bias given that PAN is temperature sensitive. " The aircraft climatology has been removed so this figure is no longer included.

"Pg 15, ln 15: " : : : may be too high : : :". If a comparison has been made then surely it either is or isn't too high for this period. " The 'may be too high' phrasing was due to the mismatch in the aircraft data climatology in terms of years of observations and the year which the model simulation represents. As mentioned the aircraft climatology has been replaced with yearly specific aircraft observations.

"Pg 15, ln 19: What is the relevance of GEOS-chem to this paper? What about the multitude of other CTM's? " This section has been removed as I no longer use the aircraft data climatology. GEOS-Chem was discussed because they used the aircraft data climatology to evaluate PAN in a similar way and there are limited papers that evaluate simulated PAN in such a manner.

"Figure 14 and 15: What is the motivation for lumping the measurements but not the model results? There are trends in surface observations related to emission trends. Is the bias larger for 2008 than 2000 i.e. does the deviance increase with time due to incorrect emission estimates?" No longer relevant, 2008 observations are now used

instead of a climatology.

---

## Author Comment (AC3) · 5 Dec 2016

"Dear authors, In my role as Executive editor of GMD, I would like to bring to your attention our Editorial version 1.1: http://www.geosci-model-dev.net/8/3487/2015/gmd-8-3487-2015.html This highlights some requirements of papers published in GMD, which is also available on the GMD website in the 'Manuscript Types' section: http://www.geoscientific-model-development.net/submission/manuscript_types.html

In particular, please note that for your paper, the following requirement has not been met in the Discussions paper: • "The main paper must give the model name and version number (or other unique identifier) in the title." For a model evaluation it is important to know, which model version exactly was evaluated. Therefore, please add

a version number for the TOMCAT model in the title upon your revised submission to GMD. Yours, Astrid Kerkweg"

Many thanks Astrid Kerkweg for bringing this to our attention. We have modified the title to include the version number v.1.6:

The TOMCAT global chemical transport model v.1.6: Description of chemical mechanism and model evaluation

---

## Author Response (AR1)

[revised manuscript text omitted]

a: Reactions are split between multiple lines.

Table 4: TOMCAT gas-phase termolecular and thermal decomposition reactions. Rate constant $k = \left(\frac{k_0[M]}{1+k_0[M]/k_\infty}\right)F_c^{(1+[log\frac{k_0[M]}{k_\infty}]^2)^{-1}}$, where $k_0$ is the low pressure limit, $k_\infty$ is the high pressure limit and $M$ is the number density in molecules/cm$^3$. $F_c = f$ when $f < 1$ else $F_c = exp(-T/f)$. Low pressure limit $k_0 = k_1 \left(\frac{T}{300}\right)^{\alpha_1} exp\left(\frac{-\beta_1}{T}\right)$ and high pressure limit $k_\infty = k_2\left(\frac{T}{300}\right)^{\alpha_2} exp\left(\frac{-\beta_2}{T}\right)$. Reaction rate references 1: Atkinson et al. (b), 2: MCM (2004), 3: Pöschl et al. (2000), 4: Atkinson et al. (c).

| | Reactants | Products | $f$ | $k_1$ | $\alpha_1$ | $\beta_1$ | $k_2$ | $\alpha_2$ | $\beta_2$ | Reference |
|---|---|---|---|---|---|---|---|---|---|---|
| 1[a] | $HO_2 + HO_2 + M$ | $\to H_2O_2 + O_2 + M$ | 0.00 | $1.90\times10^{-33}$ | 0.00 | -980.0 | $0.00\times10^{+00}$ | 0.00 | 0.0[1] | 1 |
| 2 | $HO_2 + NO_2 + M$ | $\to HO_2NO_2 + M$ | 0.60 | $1.80\times10^{-31}$ | -3.20 | 0.0 | $4.70\times10^{-12}$ | 0.00 | 0.0 | 1 |
| 3 | $HO_2NO_2 + M$ | $\to HO_2 + NO_2 + M$ | 0.60 | $4.10\times10^{-05}$ | 0.00 | 10650.0 | $4.80\times10^{+15}$ | 0.00 | 11170.0 | 1 |
| 4 | $MeCO_3 + NO_2 + M$ | $\to PAN + M$ | 0.30 | $2.70\times10^{-28}$ | -7.10 | 0.0 | $1.20\times10^{-11}$ | -0.90 | 0.0 | 1 |
| 5 | $PAN + M$ | $\to MeCO_3 + NO_2 + M$ | 0.30 | $4.90\times10^{-03}$ | 0.00 | 12100.0 | $5.40\times10^{+16}$ | 0.00 | 13830.0 | 1 |
| 6 | $N_2O_5 + M$ | $\to NO_2 + NO_3 + M$ | 0.35 | $1.30\times10^{-03}$ | -3.50 | 11000.0 | $9.70\times10^{+14}$ | 0.10 | 11080.0 | 1 |
| 7 | $NO_2 + NO_3 + M$ | $\to N_2O_5 + M$ | 0.35 | $3.60\times10^{-30}$ | -4.10 | 0.0 | $1.90\times10^{-12}$ | 0.20 | 0.0 | 1 |
| 8 | $O(^3P) + O_2 + M$ | $\to O_3 + M$ | 0.00 | $5.70\times10^{-34}$ | -2.60 | 0.0 | $0.00\times10^{+00}$ | 0.00 | 0.0 | 1 |
| 9 | $OH + NO + M$ | $\to HONO + M$ | 1420.00 | $7.40\times10^{-31}$ | -2.40 | 0.0 | $3.30\times10^{-11}$ | -0.30 | 0.0 | 1 |
| 10 | $OH + NO_2 + M$ | $\to HONO_2 + M$ | 0.40 | $3.30\times10^{-30}$ | -3.00 | 0.0 | $4.10\times10^{-11}$ | 0.00 | 0.0 | 1 |
| 11 | $OH + OH + M$ | $\to H_2O_2 + M$ | 0.50 | $6.90\times10^{-31}$ | -0.80 | 0.0 | $2.60\times10^{-11}$ | 0.00 | 0.0 | 1 |
| 12 | $EtCO_3 + NO_2 + M$ | $\to PPAN + M$ | 0.30 | $2.70\times10^{-28}$ | -7.10 | 0.0 | $1.20\times10^{-11}$ | -0.90 | 0.0 | 2 |
| 13 | $PPAN + M$ | $\to EtCO_3 + NO_2 + M$ | 0.36 | $1.70\times10^{-03}$ | 0.00 | 11280.0 | $8.30\times10^{+16}$ | 0.00 | 13940.0 | 1 |
| 14 | $MACRO_2 + NO_2 + M$ | $\to MPAN + M$ | 0.30 | $2.70\times10^{-28}$ | 0.00 | 11280.0 | $8.30\times10^{+16}$ | 0.00 | 13940.0 | 3 |
| 15 | $MPAN + M$ | $\to MACRO_2 + NO_2 + M$ | 0.30 | $4.90\times10^{-03}$ | 0.00 | 12100.0 | $5.40\times10^{+16}$ | 0.00 | 13830.0 | 3 |
| 16 | $O(^3P) + O_2 + M$ | $\to O_3 + M$ | 0.00 | $5.70\times10^{-34}$ | -2.60 | 0.0 | $0.00\times10^{+00}$ | 0.00 | 0.0 | 1 |
| 17[b] | $C_2H_4 + OH + M$ | $\to PrpeOO + M$ | 0.48 | $2.87\times10^{-29}$ | -3.10 | 0.0 | $3.00\times10^{-12}$ | -0.85 | 0.0 | 4 |
| 18[b] | $C_2H_4 + OH + M$ | $\to PrpeOO + M$ | 0.48 | $2.87\times10^{-29}$ | -3.10 | 0.0 | $3.00\times10^{-12}$ | -0.85 | 0.0 | 4 |
| 19[b] | $C_2H_4 + OH + M$ | $\to$ | 0.48 | $2.87\times10^{-29}$ | -3.10 | 0.0 | $3.00\times10^{-12}$ | -0.85 | 0.0 | 4 |
| 20 | $C_3H_6 + OH + M$ | $\to PrpeOO + M$ | 0.50 | $8.00\times10^{-27}$ | -3.50 | 0.0 | $3.00\times10^{-11}$ | -1.00 | 0.0 | 4 |

a: Reaction rate is dependent on $H_2O$ so k is weighted by factor of $1 + 1.4E - 21[H_2O]exp(2200/T)$, where [$H_2O$] is in molecules cm$^{-3}$.

b: Reactions are split between multiple lines.

Table 5: TOMCAT photolysis reactions.

| Reaction | Reactants | Products |
|---|---|---|
| 1 | $EtOOH + h\nu$ | $\rightarrow MeCHO + HO_2 + OH$ |
| 2 | $H_2O_2 + h\nu$ | $\rightarrow OH + OH$ |
| 3a | $HCHO + h\nu$ | $\rightarrow HO_2 + HO_2 + CO$ |
| 3b | $HCHO + h\nu$ | $\rightarrow H_2 + CO$ |
| 5 | $HO_2NO_2 + h\nu$ | $\rightarrow HO_2 + NO_2$ |
| 6 | $HONO_2 + h\nu$ | $\rightarrow OH + NO_2$ |
| 7a | $MeCHO + h\nu$ | $\rightarrow MeOO + HO_2 + CO$ |
| 7b | $MeCHO + h\nu$ | $\rightarrow CH4 + CO$ |
| 9 | $MeOOH + h\nu$ | $\rightarrow HO_2 + HCHO + OH$ |
| 10 | $N_2O_5 + h\nu$ | $\rightarrow NO_3 + NO_2$ |
| 11 | $NO_2 + h\nu$ | $\rightarrow NO + O(^3P)$ |
| 12a | $NO_3 + h\nu$ | $\rightarrow NO + O_2$ |
| 12b | $NO_3 + h\nu$ | $\rightarrow NO_2 + O(^3P)$ |
| 14 | $O_2 + h\nu$ | $\rightarrow O(^3P) + O(^3P)$ |
| 15a | $O_3 + h\nu$ | $\rightarrow O_2 + O(^1D)$ |
| 15b | $O_3 + h\nu$ | $\rightarrow O_2 + O(^3P)$ |
| 17 | $PAN + h\nu$ | $\rightarrow MeCO_3 + NO_2$ |
| 18 | $HONO + h\nu$ | $\rightarrow OH + NO$ |
| 19 | $EtCHO + h\nu$ | $\rightarrow EtOO + HO_2 + CO$ |
| 20 | $Me_2CO + h\nu$ | $\rightarrow MeCO_3 + MeOO$ |
| 21 | $n\text{-}PrOOH + h\nu$ | $\rightarrow EtCHO + HO_2 + OH$ |
| 22 | $i\text{-}PrOOH + h\nu$ | $\rightarrow Me_2CO + HO_2 + OH$ |
| 23 | $MeCOCH_2OOH + h\nu$ | $\rightarrow MeCO_3 + HCHO + OH$ |
| 24 | $PPAN + h\nu$ | $\rightarrow EtCO_3 + NO_2$ |
| 25 | $MeONO_2 + h\nu$ | $\rightarrow HO_2 + HCHO + NO_2$ |
| 26a | $TERPOOH + h\nu$ | $\rightarrow OH + HO_2 + MACR + MACR$ |
| 26b | $TERPOOH + h\nu$ | $\rightarrow TERPOOH + Me_2CO$ |
| 28 | $ISOOH + h\nu$ | $\rightarrow OH + MACR + HCHO + HO_2$ |
| 29 | $ISON + h\nu$ | $\rightarrow NO_2 + MACR + HCHO + HO_2$ |
| 30 | $MACR + h\nu$ | $\rightarrow MeCO_3 + HCHO + CO + HO_2$ |
| 31 | $MPAN + h\nu$ | $\rightarrow MACRO_2 + NO_2$ |
| 32a | $MACROOH + h\nu$ | $\rightarrow OH + HO_2 + OH + HO_2$ |
| 32b | $MACROOH + h\nu$ | $\rightarrow HACET + CO + MGLY + HCHO$ |
| 34 | $HACET + h\nu$ | $\rightarrow MeCO_3 + HCHO + HO_2$ |
| 35 | $MGLY + h\nu$ | $\rightarrow MeCO_3 + CO + HO_2 +$ |
| 36 | $NALD + h\nu$ | $\rightarrow HCHO + CO + NO_2 + HO_2$ |
| 37 | $MeCO_3H + h\nu$ | $\rightarrow MeOO + OH$ |
| 38a | $O_3S + h\nu$ | $\rightarrow O_2 + O(1D)S$ |
| 38b | $O_3S + h\nu$ | $\rightarrow O_2 + O(3P)S$ |
| 40a | $BtOOH + h\nu$ | $\rightarrow MEK + MEK + EtOO + MeCHO$ |

| Reaction | Reactants | Products |
|----------|-----------|----------|
| 40b | BtOOH + h$\nu$ | $\rightarrow HO_2 + HO_2$ |
| 40c | BtOOH + h$\nu$ | $\rightarrow OH + OH + OH$ |
| 43 | MEK + h$\nu$ | $\rightarrow MeCO_3 + EtOO$ |
| 44 | MeCOCOMe + h$\nu$ | $\rightarrow MeCO_3 + MeCO_3$ |
| 45 | MEKOOH + h$\nu$ | $\rightarrow MeCO_3 + MeCHO + OH$ |
| 46a | ONIT + h$\nu$ | $\rightarrow NO_2 + MEK + HO_2 + EtOO$ |
| 46b | ONIT + h$\nu$ | $\rightarrow MeCHO + ONIT$ |
| 48a | AROMOOH + h$\nu$ | $\rightarrow OH + Me_2CO + HO_2 + CO$ |
| 48b | AROMOOH + h$\nu$ | $\rightarrow MeCO_3 + AROMOOH$ |

1090

**Table 6.** List of $\gamma$ values used in TOMCAT for heterogeneous uptake of $N_2O_5$ by aerosol.

| Aerosol Type | Reaction Probability |
|---|---|
| | (T=temperature (K), RH= relative humidity (%)) |
| Sulphate | $\gamma = \alpha \times 10^{\beta}$ |
| | $\alpha = 2.79 \times 10^{-4} + 1.3$ |
| | $\times 10^{-4} \times RH - 3.43$ |
| | $\times 10^{-6} \times RH^2 + 7.52$ |
| | $\times 10^{-8} \times RH^3$ |
| | $\beta = 4 \times 10^{-2} \times (T-294) \ (T \geq 282K)$ |
| | $\beta = -0.48 \ (T < 282K)$ |
| Organic Carbon | $\gamma = RH \times 5.2 \times 10^{-4} \ (RH < 57\%)$ |
| Black Carbon | $\gamma = 0.005$ |
| Sea Salt | $\gamma = 0.005 \ (RH < 62\%)$ |
| | $\gamma = 0.03 \ (RH \geq 62\%)$ |
| Dust | $\gamma = 0.02$ |

**Table 7.** Model diagnostics compared to previously published values

| Diagnostic | TOMCAT | Published Values | Reference |
|---|---|---|---|
| $O_3$ Burden (Tg)[a] | 331 | 337±23 | Young et al. (2013) |
| OH concentration | 1.08 | 0.94–1.06 | Krol and Lelieveld (2003) , Prinn et al. (2001) , |
| ($\times 10^6$ molecules/cm$^3$)[b] | | | Bousquet et al. (2005) , Wang et al. (2008) |
| $CH_4$ lifetime (yrs) | 7.9 | 9.3±.0.9 | Voulgarakis et al. (2013) |

a: Annual mean; b: Mass-weighted annual mean.

---

## Referee Report (RR1)

**A review report on "The TOMCAT global chemical transport model v1.6: Description of chemical mechanism and model evaluation" by Monks et al., 2017.**

**General**

The study presents a comprehensive description and validation of a tropospheric chemical mechanism currently used in the TOMCAT global chemical transport model. The current description is an update to the original scheme of Chipperfield et al. (2006) and references therein. However, in view of recent scientific updates in chemical mechanisms addressing isoprene recycling under low $NO_x$ conditions as well as $RO_2/HO_2/ROOH$ formation in remote regions in the EMAC (Lelieved et al., 2016), the MCMv3.3.1 (Jenkin et al, 2016) and GEOS-Chem (Fisher et al., 2016), my main concern here is what are the scientific benefits of such (updated) chemical mechanism, which does not cover any of these known biases. Also, this updated chemical scheme is unfortunately based on outdated chemical scheme (MIM), which was replaced by MIM2 and most recently by MIM3 (Lelieveld et al., 2016). The authors are also using the most outdated version of MCMv3.1 instead of the most recent versions of MCMv3.3 and MCMv3.3.1? I think that this is a serious issue since the authors are aiming to document and publish a reduced chemical mechanism that is literally based on outdated schemes..?

It is certainly important to document a chemical scheme that is used for scientific research, but only if there are important updates that warrant publication, **compared to the original scheme**, which is not the case here, at least not demonstrated. The study does present a comprehensive validation of the chemistry scheme but without a sufficient justification for the scientific benefits, especially given the recent updates in isoprene and $HO_x$ chemistry in the last few years (see above), none of which are implemented here. I wished also to see a long simulation experiment (e.g, 20 years) to see if the model can capture, e.g., methane trend and growth rates, CO and $O_3$ inter annual variability, especially given the large biases in OH, CO, $O_3$ and VOCs.

**Some Specific Comments**

**P1 L 10-12:** Authors may also show the mean global OH concentration along with the range of multimodal ACCMIP values and MCF-based estimates.

**P1 L13:** Would this bias result from using a biased/different water vapor verticals profiles, photolysis profiles?

**P4 L 7-10:** It is unfortunate that the authors decided to use the old MIM version (Pöschl et al., 2000), which has been updated several times, with most recent versions MIM2 (Taraborelli et al., 2012) and MIM3 (Lelieveld et al., 2016) addressing important update in Isoprene and $HO_x$ chemistry.

**P4 L13-14:** Since the authors aim at documenting this scheme, could the authors further elaborate why does the extended chemistry increase the burdens of the mentioned species? Would be also very interesting to see some scientific discussions and justifications compared to the original scheme.

**P4 L24:** Again, The users are using the most outdated version of MCMv3.1 instead of the most recent versions of MCMv3.3 and MCMv3.3.1? I think that is

a serious issue since the authors are aiming to publish a chemical mechanism that is literally based on outdated schemes?

**P6 L3:** Change "cheap" to "efficient". Is there is any quantitative analysis that support the author's claim that the option "TOMCAT-GLOMAP" is computationally efficient? Compared to what?

**P9 L 12:** Fig. 3, Why OH levels are very high over the sea southeast and southwest of India?

OH vertical profile does not seem typical, **e.**g., compared to Spivakovsky et al. (2000). Although the authors mentioned this earlier but some discussions are needed here to address this difference?

**P10 L7:** OH from TOMCAT is still too high in the Arabian Sea to the coast of India, not related to ship traffic as in Voulgarakis et al. (2013), artifacts?

**P11 L5-15**: Although the authors discussed the higher OH levels near the surface as opposed to other models, the issue still not corrected, which, as the authors mentioned, affect e.g., model calculation of methane lifetime. I think whether this issue is related to the driven $H_2O$ profiles, photolysis rate calculations, or the underestimated anthropogenic emissions (CO is a main OH sink near the surface), it has to be corrected, otherwise, how this new scheme can be used for comparison with other models, which see high OH near 600 hpa for know reasons (see e.g., Spivakovsky et al., 2000).

**References**

Fisher, J. A., Jacob, D. J., Travis, K. R., Kim, P. S., Marais, E. A., Chan Miller, C., Yu, K., Zhu, L., Yantosca, R. M., Sulprizio, M. P., Mao, J., Wennberg, P. O., Crounse, J. D., Teng, A. P., Nguyen, T. B., St. Clair, J. M., Cohen, R. C., Romer, P., Nault, B. A., Wooldridge, P. J., Jimenez, J. L., Campuzano-Jost, P., Day, D. A., Hu, W., Shepson, P. B., Xiong, F., Blake, D. R., Goldstein, A. H., Misztal, P. K., Hanisco, T. F., Wolfe, G. M., Ryerson, T. B., Wisthaler, A., and Mikoviny, T.: Organic nitrate chemistry and its implications for nitrogen budgets in an isoprene- and monoterpene-rich atmosphere: constraints from aircraft (SEAC[4]RS) and ground-based (SOAS) observations in the Southeast US, Atmos. Chem. Phys., 16, 5969-5991, doi:10.5194/acp-16-5969-2016, 2016.

Lelieveld, J., Gromov, S., Pozzer, A., and Taraborrelli, D.: Global tropospheric hydroxyl distribution, budget and reactivity, Atmos. Chem. Phys., 16, 12477-12493, doi:10.5194/acp-16-12477-2016, 2016.

Spivakovsky, C. M., Logan, J. A., Montzka, S. A., Balkanski, Y. J., Foreman-Fowler, M., Jones, D. B. A., Horowitz, L.W., Fusco, A. C., Brenninkmeijer, C. A. M., Prather, M. J., Wofsy, S. C., and McElroy, M. B.: Three-dimensional climatological distribution of tropospheric OH: Update and evaluation, J. Geophys. Res., 105, 8931–8980, doi:10.1029/1999JD901006, 2000.

Taraborrelli, D., Lawrence, M. G., Crowley, J. N., Dillon, T. J., Gromov, S., Groß, C. B. M., Vereecken, L., and Lelieveld, J.: Hydroxyl radical buffered by isoprene oxidation over tropical forests, Nat. Geosci., 5, 190–193, 2012.